# A prebiotically plausible scenario of an RNA–peptide world

Felix Müller[1,2], Luis Escobar[1,2], Felix Xu[1], Ewa Węgrzyn[1], Milda Nainytė[1], Tynchtyk Amatov[1], Chun-Yin Chan[1], Alexander Pichler[1] & Thomas Carell[1✉]

The RNA world concept[1] is one of the most fundamental pillars of the origin of life theory[2–4]. It predicts that life evolved from increasingly complex self-replicating RNA molecules[1,2,4]. The question of how this RNA world then advanced to the next stage, in which proteins became the catalysts of life and RNA reduced its function predominantly to information storage, is one of the most mysterious chicken-and-egg conundrums in evolution[3–5]. Here we show that non-canonical RNA bases, which are found today in transfer and ribosomal RNAs[6,7], and which are considered to be relics of the RNA world[8–12], are able to establish peptide synthesis directly on RNA. The discovered chemistry creates complex peptide-decorated RNA chimeric molecules, which suggests the early existence of an RNA–peptide world[13] from which ribosomal peptide synthesis[14] may have emerged[15,16]. The ability to grow peptides on RNA with the help of non-canonical vestige nucleosides offers the possibility of an early co-evolution of covalently connected RNAs and peptides[13,17,18], which then could have dissociated at a higher level of sophistication to create the dualistic nucleic acid–protein world that is the hallmark of all life on Earth.

A central commonality of all cellular life is the translational process, in which ribosomal RNA (rRNA) catalyses peptide formation with the help of transfer RNAs (tRNA), which function as amino acid carrying adapter molecules[14,19,20]. Comparative genomics[21] suggests that ribosomal translation is one of the oldest evolutionary processes[15,16,22,23], which dates back to the hypothetical RNA world[1–4]. The questions of how and when RNA learned to instruct peptide synthesis is one of the grand unsolved challenges in prebiotic evolutionary research[3–5].

The immense complexity of ribosomal translation[14] demands a step-wise evolutionary process[11]. From the perspective of the RNA world, at some point RNA must have gained the ability to instruct and catalyse the synthesis of, initially, just small peptides. This initiated the transition from a pure RNA world[1] into an RNA–peptide world[13]. In this RNA–peptide world, both molecular species could have co-evolved to gain increasing 'translation' and 'replication' efficiency[17].

To gain insight into the initial processes that may have enabled the emergence of an RNA–peptide world[13], we analysed the chemical properties of non-canonical nucleosides[6,7], which can be traced back to the last universal common ancestor and, as such, are considered to be 'living molecular fossils' of an early RNA world[8–12].

This approach, which can be called 'palaeochemistry', enabled us to learn about the chemical possibilities that existed in the RNA world and, therefore, sets the chemical framework for the emergence of life. In contrast to earlier investigations of the origin of translation[24–29], we used naturally occurring non-canonical vestige nucleosides and conditions compatible with aqueous wet–dry cycles[30,31].

## Peptide synthesis on RNA

In modern tRNAs (Fig. 1a), the amino acids that give peptides are linked to the CCA 3′ terminus via a labile ester group[32]. Some tRNAs, however, contain additional amino acids in the form of amino acid-modified nucleosides, for example, g⁶A (ref. [33]), t⁶A (ref. [34]) and m⁶t⁶A (ref. [35]), which are found directly next to the anticodon loop at position 37. Other non-canonical vestige nucleosides often present in the wobble position 34 are nm⁵U and mnm⁵U (refs. [36–38]).

Close inspection of their chemical structures (Fig. 1b) suggests that if they are in close proximity (step 1), an RNA-based peptide synthesis may be able to start (step 2), which would create, via a hairpin-type intermediate, a peptide attached by a urea linkage to the nucleobase (m⁶)aa⁶A. Cleavage of the urea[39,40] (step 3) would furnish RNA with a peptide connected to a (m)nm⁵U (step 4). Subsequently, strand displacement with a new (m⁶)aa⁶A strand may finally enable the next peptide elongation step.

To investigate the potential evolution of an RNA–peptide world, we synthesized two complementary sets of RNA strands, **1a**–**1j** and **2a**–**2c** (Fig. 2). The first set contained various m⁶aa⁶A nucleotides[41] at the 5′ end (**1a**–**1j**) as RNA donor strands. The complementary RNA acceptor strands were prepared with an (m)nm⁵U nucleotide at the 3′ terminus (**2a**–**2c**). Figure 2a shows the reactions between **1a** and **2a**. The analytical data are presented in Fig. 2b. We hybridized **1a** with **2a** and activated the carboxylic acid of **1a** using reagents such as EDC[42]/Sulfo-NHS[43], DMTMM·Cl[43] or methyl isonitrile[44] (pH 6, 25 °C). In all cases we observed high yielding product formation (Fig. 2c).

A kinetic analysis shows that the nature of the amino acid affects the coupling rate (Fig. 2d). For example, G (in **1a**) couples to **2c** with an apparent rate constant ($k_{app}$) of 0.1 h⁻¹. For the amino acids L (in **1d**), T (in **1e**) and M (in **1h**) a fourfold higher rate constant (≈0.4 h⁻¹) was determined, and the highest rate was measured for F (in **1g**) with $k_{app}$ > 1 h⁻¹. These differences establish a pronounced amino acid selectivity in the coupling reaction, probably as a result of distinct pre-organizations. We next reduced the length of the RNA donor strand to five, and finally to three, nucleotides (Supplementary Information). We detected coupling even with a trimer

[1]Department of Chemistry, Ludwig-Maximilians-Universität (LMU) München, Munich, Germany. [2]These authors contributed equally: Felix Müller, Luis Escobar. ✉e-mail: thomas.carell@lmu.de

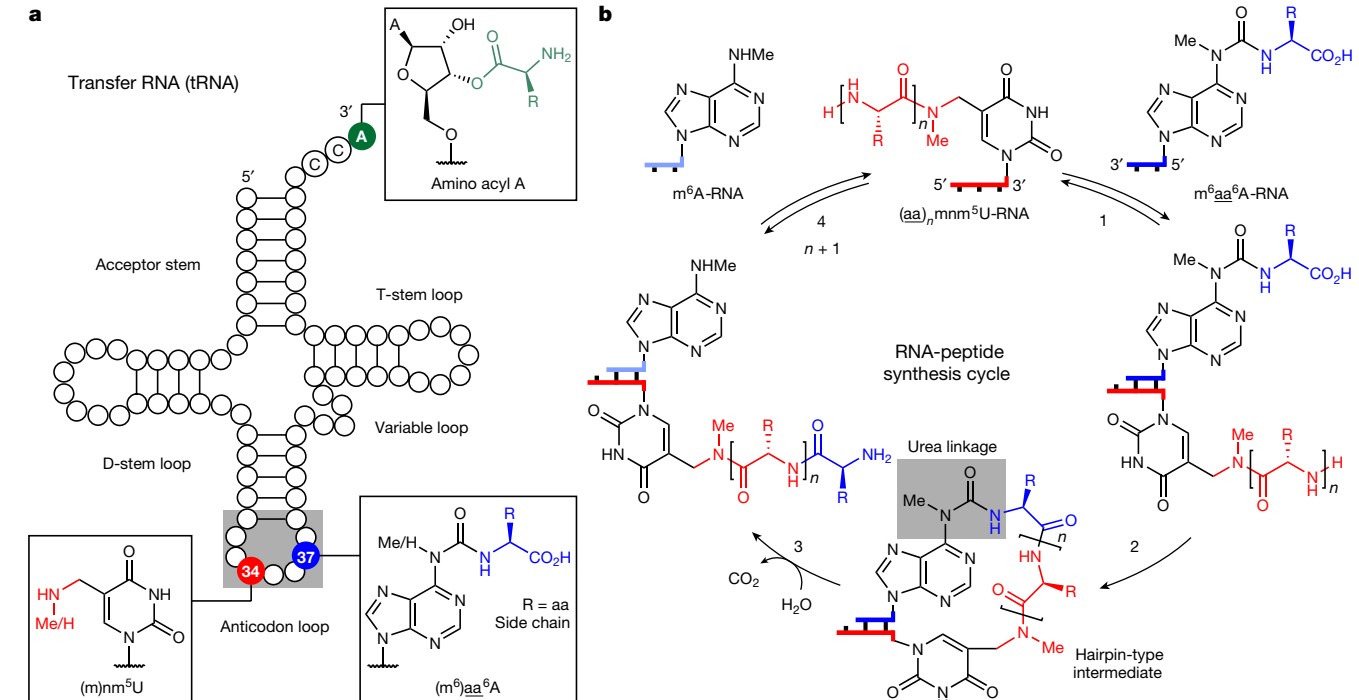

**Fig. 1 | Concept of how nucleoside relics of the RNA world enable RNA-based peptide synthesis. a**, tRNA structure showing selected ribose and nucleobase modifications. The 3'-amino acid-acylated adenosine is located at the CCA 3' end in contemporary tRNAs. 5-Methylaminomethyl uridine, mnm⁵U, is found in the wobble position 34. The amino acid-modified carbamoyl adenosine, (m⁶)aa⁶A (aa, amino acid), is present at position 37 in certain tRNAs. **b**, General RNA–peptide synthesis cycle based on mnm⁵U and m⁶aa⁶A. The structures of oligonucleotides are simplified and only terminal nucleobases are drawn.

RNA donor strand, although it required duplex-enforcing high salt and low temperature conditions (1 M NaCl and 0 °C). The interaction of three nucleotides on the donor with the corresponding triplet on the acceptor seems to be the lower limit for productive coupling. Interestingly, this is the size of the codon–anticodon interaction in contemporary translation[11,18].

We next investigated coupling of the nitrile derivative of **1a** (m⁶g$_{CN}$⁶A, **1j**) with the different acceptors **2a**–**2c** under the recently described prebiotically plausible thiol activation conditions[45] (DTT, pH 8, 25 °C). Here also, the coupling products were obtained within a few hours (Fig. 2c). For example, the combination of nm⁵U **2b** with **1a** gives coupling yields of 64% and 66% using EDC/Sulfo-NHS or DMTMM·Cl, respectively. Coupling of **1a** and **2a**, featuring a secondary amine, afforded **3a** in 16% and 33% yields. The nitrile of **1j** afforded yields of up to 65% after thiol activation coupling.

We next measured the stability of the hairpin-type intermediates. For the hairpin **3a** (Fig. 2a), a melting temperature ($T_m$) of approximately 87 °C was determined, which in comparison to the starting duplex (approximately 30 °C for **1a·2a**, see Supplementary Information), proves that the peptide formation reaction generated thermally more stable structures. This could have been an advantage during wet–dry cycling under early Earth conditions.

The discovered concept also enabled the synthesis of longer peptides. When we used 3'-vmnm⁵U-RNA-5' **2c** as the acceptor, we observed, on reaction with **1a**–**1j**, peptide bond formation with up to 77% yield (Fig. 2c, d and Fig. 3a).

We next studied the cleavage of the urea linkage and found that this reaction was possible at elevated temperatures (90 °C) in water at pH 6 (Fig. 2a, b). After 6 h, the products, m⁶A-containing RNA **4** and RNA **5a** were formed already with a yield of 15%.

## Longer peptide structures on RNA

We next investigated how the length of the generated peptides influences the coupling reaction (Fig. 3 and Extended Data Fig. 1). For this study we used synthetic 3'-peptide-mnm⁵U-RNA-5' acceptor strands as starting materials (Supplementary Information). The synthesized acceptor strands were hybridized to the donor strand **1a**. After carboxylic acid activation, rapid formation of elongated hairpin-type intermediates with yields between 40% and 60% was observed (Fig. 3b). We found that the coupling yields did not drop substantially with increasing peptide length, suggesting that other factors, such as the RNA hybridization kinetics, are rate limiting. In all cases, the subsequent urea cleavage (pH 4, 90 °C) affords dipeptide- to hexapeptide-decorated RNAs in 10–15% yield. These modest yields are the result of substantial RNA degradation, driven by the pH and temperature conditions that were used. The decomposition of RNA, however, can be overcome by using 2'-OMe nucleotides (see 'Stepwise growth of peptides on RNA'), which are also vestiges of the early RNA world[46].

During urea cleavage we detected competing formation of hydantoin side products[47], depending on the pH and temperature (Fig. 3a). Under mildly acidic conditions (pH 6, 90 °C), exclusive formation of the hydantoin product, cyclic-**5c**, was observed. Reducing the temperature and a shift to higher acidity (pH 4, 60 °C) led to the preferential formation of the peptide product, **5c** (approximately 7:1 **5c**:cyclic-**5c** ratio).

## Fragment coupling on RNA

We investigated whether longer peptides can also be generated by fragment coupling chemistry with RNA donor strands containing an already longer peptide (m⁶peptide⁶A). This is essential because an RNA–peptide world, with initially low chemical efficiency, might have been limited to the synthesis of smaller peptides. We found that the required adenosine nucleosides, containing a whole peptide attached to the $N^6$-position, are available if the peptides that are produced by RNA degradation of the RNA–peptide chimeras, for example, can react with nitrosated $N^6$-methylurea adenosine (Fig. 4a). When we treated $N^6$-methylurea adenosine with NaNO₂ (5% H₃PO₄) and added the solution to triglycine (pH 9.5), we obtained the peptide-coupled adenosine nucleoside ggg⁶A

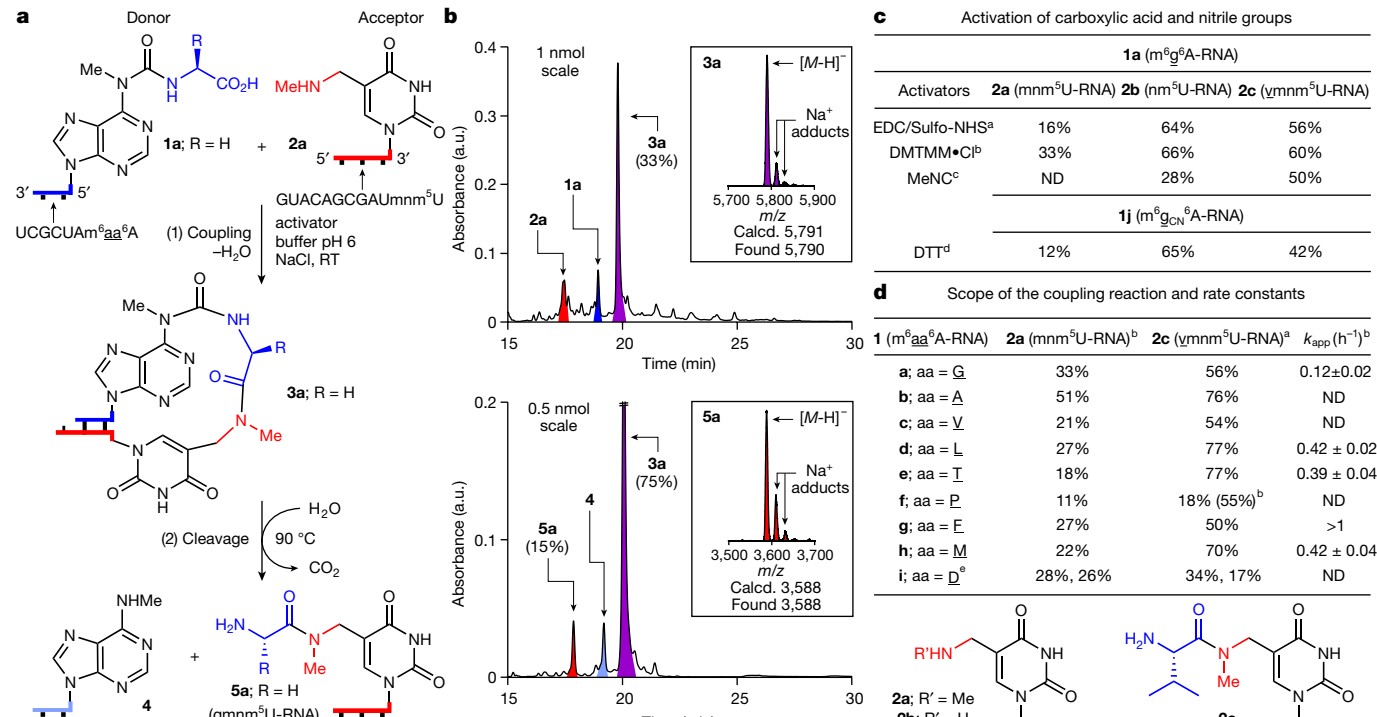

**Fig. 2 | Peptide synthesis on RNA with terminal (m)nm⁵U and m⁶aa⁶A nucleotides. a**, Reaction scheme for **1a** (5′-m⁶g⁶A-RNA-3′) and **2a** (3′-mnm⁵U-RNA-5′) with coupling (1) and cleavage (2). **b**, HPLC chromatograms of the crude reaction mixtures, obtained after coupling of **1a** with **2a** using DMTMM·Cl (see reaction condition b) and cleavage of **3a** (100 mM MES buffer pH 6, 100 mM NaCl, 90 °C, 6 h). HPLC peaks of RNAs are coloured: donor in blue; acceptor in red; hairpin-type intermediate in purple; and cleaved donor strand in pale blue. The insets show MALDI-TOF data (negative mode) of the isolated products **3a** and **5a**. Calcd., calculated. **c**, Coupling results obtained with different activators for **1a** and **1j** with **2a**–**2c**. **d**, Coupling reactions with

different donors **1a**–**1i** and acceptors **2a**, **2c**, and apparent rate constants ($k_{app}$) of selected coupling reactions with **2c**. All coupling reactions were carried out using a concentration of 50 μM for **1a**–**1j** and 50 μM for **2a**–**2c** (100 mM NaCl, 25 °C). ᵃ50 mM EDC/Sulfo-NHS (100 mM MES buffer pH 6, 24 h). ᵇ50 mM DMTMM·Cl (100 mM MES buffer pH 6, 24 h). ᶜ50 mM MeNC (50 mM DCI buffer pH 6, 5 days). ᵈ50 mM DTT (100 mM borate buffer pH 8, 24 h). ᵉThe two yields with **1i** (aa, D) describe the reaction of the aspartic acid α-COOH and of the side chain COOH. An assignment was not performed. RT, room temperature; ND, not determined.

---

in approximately 65% yield. Incorporation of (m⁶)ggg⁶A into RNA and hybridization of this donor strand with a 3′-ggvmnm⁵U-RNA-5′ acceptor strand furnished, after coupling and urea cleavage, the RNA–peptide chimera 3′-gggggvmnm⁵U-RNA-5′ (53% coupling, approximately 10% cleavage; Fig. 4b, left). We could also directly transfer longer peptides. When we hybridized the 5′-m⁶gaggg⁶A-RNA-3′ donor with the 3′-agggvmnm⁵U-RNA-5′ acceptor, 3′-gagggagggvmnm⁵U-RNA-5′ was obtained as the product (56% coupling, approximately 9% cleavage; Fig. 4b, right). These experiments suggest the possibility of generating highly complex RNA–peptide chimeras with just a small number of reaction steps[48].

## Multiple peptide growth on RNA

We next investigated whether peptide growth is possible at different RNA positions simultaneously. To this end, we examined the simultaneous binding of different donor strands to one or two acceptor strands. We hybridized two donor strands (7-mer: 5′-m⁶g⁶A-RNA-3′ and 10-mer: 5′-m⁶v⁶A-RNA-3′) to a single RNA acceptor strand (21-mer) with a central gmnm⁵U and a 3′ terminal nm⁵U (Fig. 5a, left). On activation of the carboxylic acids, a GG-dipeptide was synthesized in the centre of the RNA, whereas a valine amino acid was attached to the 3′ end of the acceptor strand. In a different experiment, we hybridized an RNA donor strand (22-mer), containing both a 3′-m⁶g⁶A and a 5′-m⁶v⁶A, to two different acceptor RNAs, containing a central vmnm⁵U (21-mer) and a 3′ terminal vmnm⁵U (11-mer) (Fig. 5a, right). On activation, we observed formation of a central GV- and a terminal VV-dipeptide.

## Effect of base pairing

To investigate the importance of sequence complementarity, we added two RNA donor strands of different lengths (7-mer: 5′-m⁶g⁶A-RNA-3′ and 11-mer: 5′-m⁶v⁶A-RNA-3′) to an acceptor strand with a vmnm⁵U at the 3′ end (11-mer: **2c**) (Fig. 5b, left). On the basis of the melting temperatures of the two possible duplexes (approximately 30 °C for the 7-mer·11-mer and 59 °C for the 11-mer·11-mer, see Supplementary Information), only formation of the VV-dipeptide RNA conjugate, derived from the thermodynamically more stable duplex, was observed. Finally, we mixed two RNA donor strands of identical length (7-mer). The first contained a 5′-m⁶l⁶A and the second a 5′-m⁶g⁶A, together with two mismatches. We added this mixture to an RNA acceptor strand (11-mer: **2c**) with a 3′-vmnm⁵U nucleotide (Fig. 5b, right). In this experiment, exclusive formation of the LV-dipeptide was found, generated from the fully complementary strands and thus the more stable duplex. Collectively, these results support that full complementarity is needed for efficient peptide synthesis.

## Stepwise growth of peptides on RNA

We finally investigated whether one-pot stepwise growth of a peptide on RNA is possible (Fig. 5c). To increase the stability of the RNA towards phosphodiester hydrolysis, as needed for this experiment, we used the RNA acceptor strand **2g**, in which the contemporary canonical bases were replaced by the non-canonical 2′-OMe nucleotides: $A_m$, $C_m$, $G_m$ and $U_m$. The strand **2g** was equipped with an additional 3′-mnm⁵U

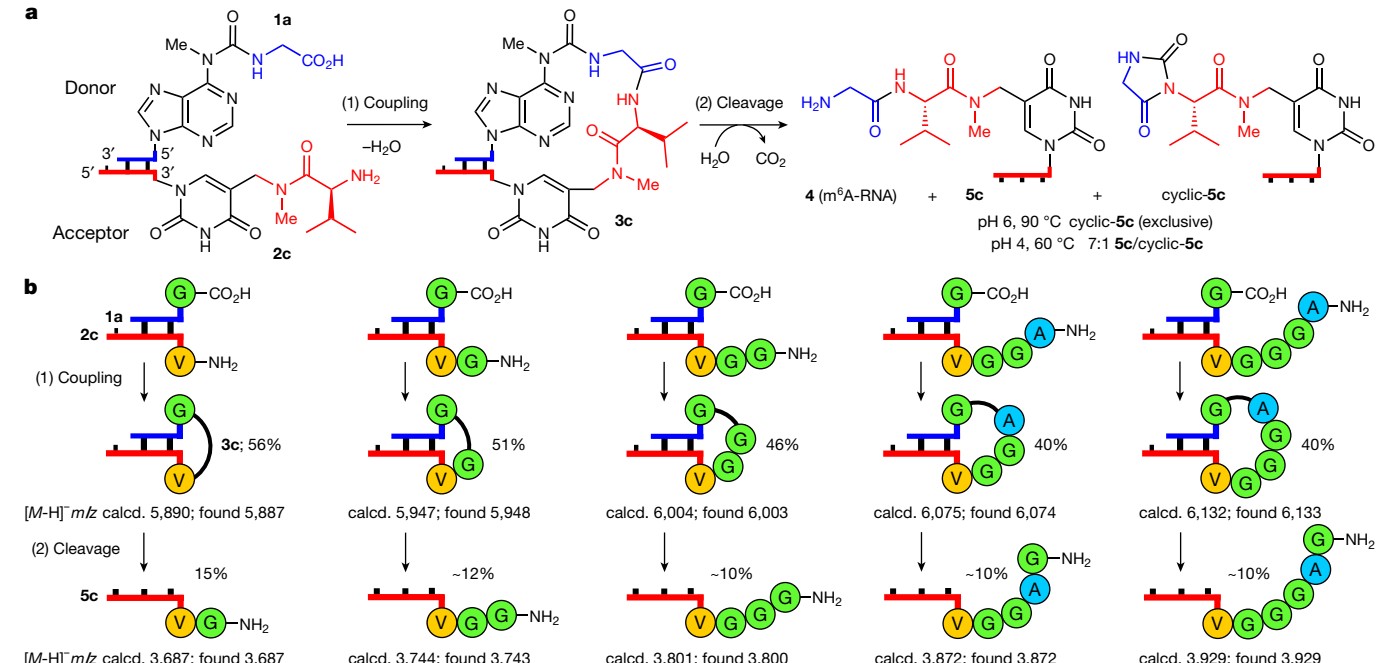

**Fig. 3 | Growth of longer peptide structures on RNA. a**, Scheme for the reaction of **1a** (5′-m⁶g⁶A-RNA-3′) with **2c** (3′-vmnm⁵U-RNA-5′) including coupling (1) and cleavage (2). **b**, Coupling reactions between **1a** and RNA–peptide acceptor strands using EDC/Sulfo-NHS (see reaction condition a in Fig. 2) and cleavage reactions of the coupled compounds (100 mM acetate buffer pH 4, 100 mM NaCl, 90 °C, 6 h). MALDI-TOF data (negative mode) of the isolated products are given.

nucleotide. For the experiment we used the same amount of donor strand for all coupling steps and performed filtration steps to remove remaining activator. After two couplings, two urea cleavages and two filtrations, we observed, by high-performance liquid chromatography (HPLC) analysis, the presence of the product 3′-ggmnm⁵U-RNA-5′ **7g** (Fig. 5c, left). The circumvented material consuming isolation steps

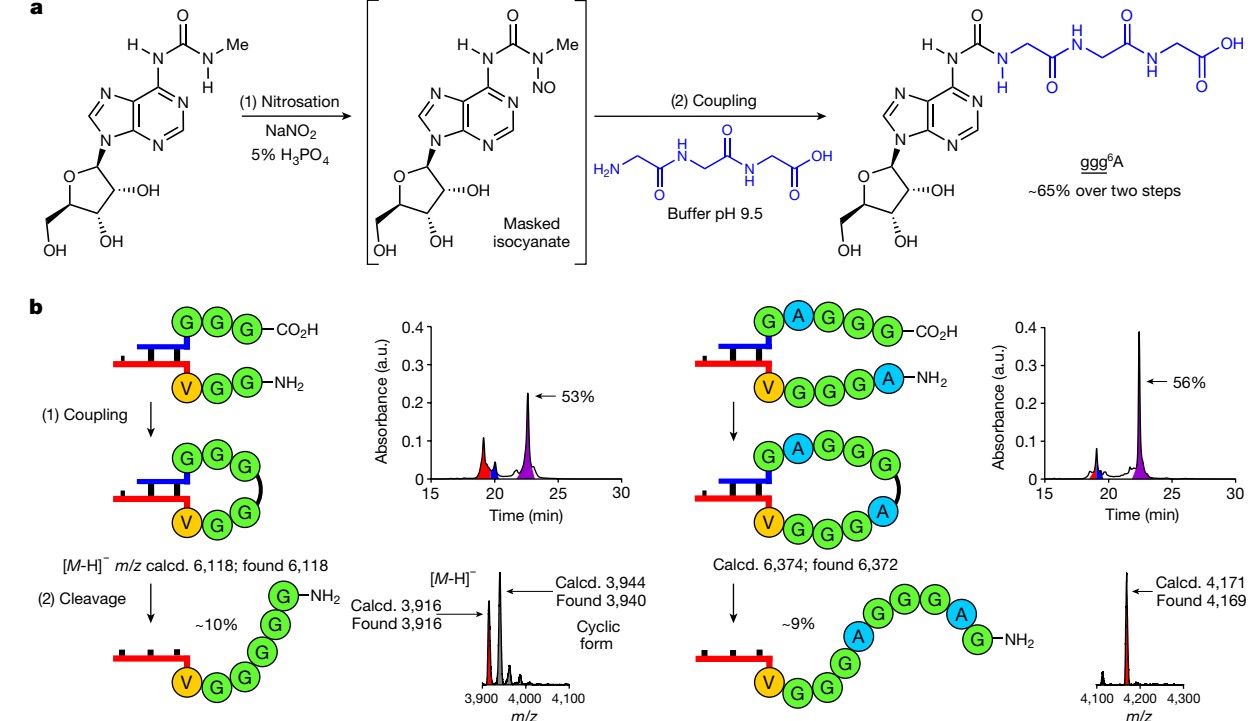

**Fig. 4 | Capture of peptides by nitrosated N⁶-methylurea adenosine for fragment condensation. a**, Prebiotically plausible formation of peptide⁶A structures, such as ggg⁶A. **b**, Coupling reactions between RNA-peptide conjugates using EDC/Sulfo-NHS (see reaction condition a in Fig. 2) and cleavage reactions of the coupled compounds (see reaction conditions in

Fig. 3). HPLC chromatograms show the crude mixtures of the coupling reactions. The RNA signals are coloured: donor in blue; acceptor in red; and hairpin-type intermediate in purple. MALDI-TOF data (negative mode) are shown for the isolated products, together with the 5′-m⁶A-RNA-3′ strand **4** and the hydantoin side product (cyclic form) in the case indicated.

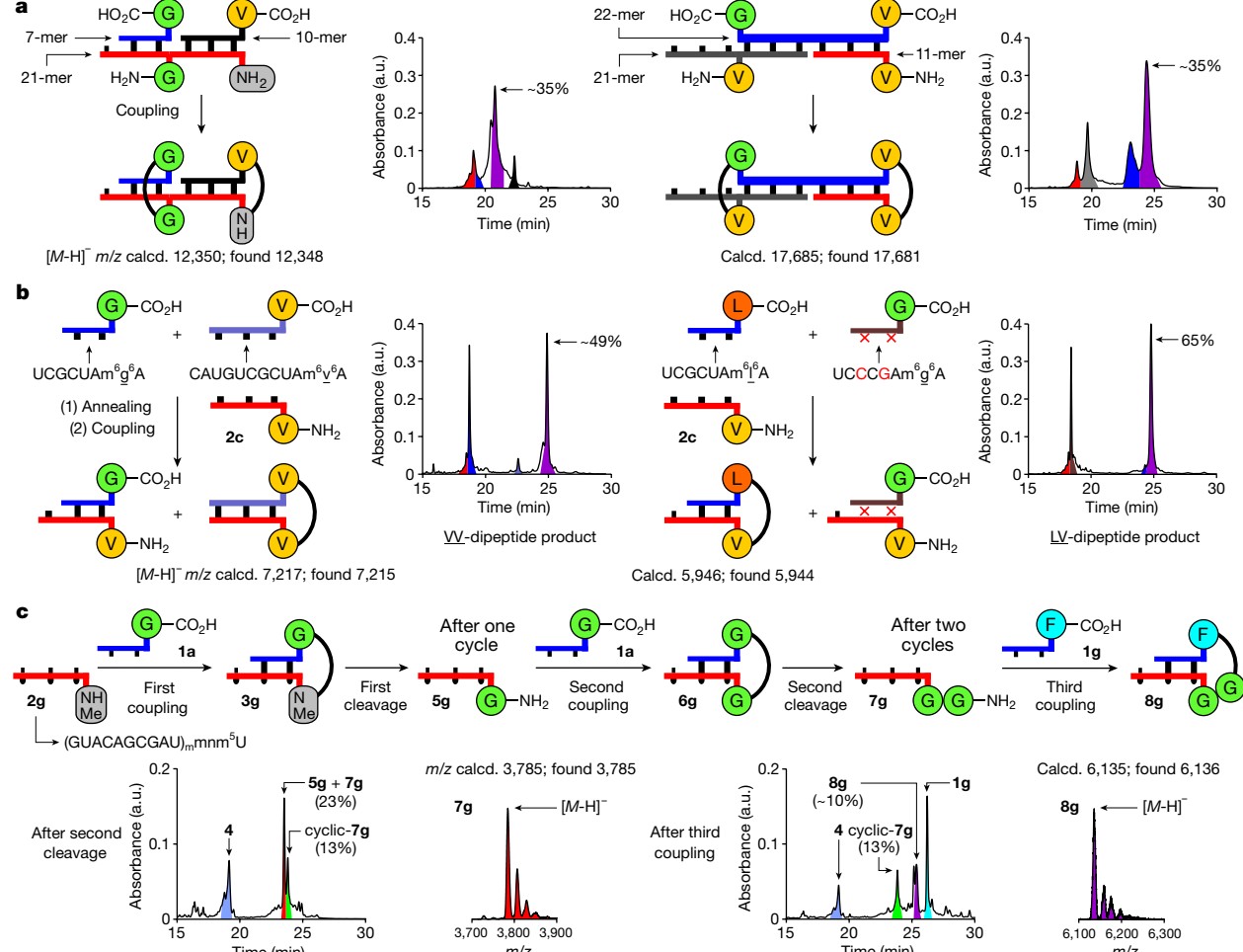

**Fig. 5 | Parallel growth of peptides at various positions on RNA, effect of base pairing and RNA–peptide synthesis cycles. a**, Coupling of oligonucleotides containing multiple donor or acceptor units (EDC/Sulfo-NHS, see reaction condition a in Fig. 2). **b**, Annealing followed by coupling (EDC/Sulfo-NHS, see reaction condition a in Fig. 2) of an acceptor strand with donor strands of different length (left) or base sequence (right). **c**, Two RNA–peptide synthesis cycles with a third coupling step using a 2′-OMe acceptor strand and performed under one-pot conditions with intermediary filtration to remove the remaining activator (coupling: DMTMM·Cl, see reaction condition b in Fig. 2; cleavage: 100 mM acetate buffer pH 4, 100 mM NaCl, 90 °C, 24 h; MES buffer pH 6 was used in the first cleavage reaction). HPLC chromatograms show the crude mixtures of the coupling and cleavage reactions. Peaks of RNA strands are coloured as in the reaction scheme. MALDI-TOF data (negative mode) of the isolated products are given.

(Extended Data Fig. 2) enabled us to obtain the product in an overall yield of about 18%. A final, third coupling reaction with the 5′-m⁶f⁶A donor strand **1g** furnished the FGG-hairpin intermediate **8g** in approximately 10% overall yield (Fig. 5c, right).

We next studied fragment condensation with the 5′-m⁶ggg⁶A-RNA-3′ donor strand and the complementary 3′-aggmnm⁵U-RNA-5′ acceptor strand, consisting only of 2′-OMe nucleotides. Here, coupling with approximately 50% and urea cleavage with approximately 85% generated the product 3′-gggaggmnm⁵U-RNA-5′, together with some of the hydantoin side product (Supplementary Information). Together these data show that, with the help of 2′-OMe nucleotides, peptides can grow on RNA in a stepwise fashion and via fragment condensation to generate higher complexity.

## Discussion

The plausible formation of catalytically competent and self-replicating RNA structures without the aid of proteins is one of the major challenges for the model of the RNA world[1–4]. It is difficult to imagine how an RNA world with complex RNA molecules could have emerged without the help of proteins and it is hard to envision how such an RNA world transitions into the modern dualistic RNA and protein world, in which RNA predominantly encodes information whereas proteins are the key catalysts of life.

We found that non-canonical vestige nucleosides[8–12], which are key components of contemporary RNAs[6,7], are able to equip RNA with the ability to self-decorate with peptides. This creates chimeric structures, in which both chemical entities can co-evolve in a covalently connected form[13], generating gradually more and more sophisticated and complex RNA–peptide structures. Although, in this study, we observe peptide coupling on RNA in good yields, the efficiency will certainly improve if we allow optimization of the structures and sequences of the RNA–peptides by chemical evolution. The simultaneous presence of the chemical functionalities of RNA and amino acids certainly increases the chance of generating catalytically competent structures. The stabilization of RNA by incorporation of 2′-OMe nucleotides significantly improved the urea cleavage yield.

Interestingly, in the coupling step we observed large differences in the rate constants, which suggests that our system has the potential to preferentially generate certain peptides. We also found that peptides can simultaneously grow at multiple sites on RNA on the basis of rules determined by sequence complementarity, which is the indispensable requirement for efficient peptide growth.

All these data together support the idea that non-canonical vestige nucleosides in RNA have the potential to create peptide self-decorating RNAs and hence an RNA–peptide world. The formed RNA–peptide chimeras are comparatively stable, and so it is conceivable that some of these structures learned, at some point, to activate amino acids by adenylation[49] and to transfer them onto the ribose OH groups[50] to capture the reactivity in structures that were large and hydrophobic enough to exclude water. This would then have been the transition from the non-canonical nucleoside-based RNA–peptide world to the ribosome-centred translational process that is a hallmark of all life on Earth today.

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

## Methods

### General method for the peptide coupling reactions

The RNA donor and acceptor strands (1:1 ratio, 5 nmol of each strand) were annealed with NaCl (5 μl from a 1 M aqueous solution) by heating at 95 °C for 4 min, followed by cooling down slowly to room temperature. After that, MES buffer pH 6 (25 μl from a 400 mM aqueous solution) and NaCl (5 μl from a 1 M aqueous solution) were added to the oligonucleotide solution. Finally, carboxylic acid or nitrile activator/s (10 μl of each component from a 500 mM aqueous solution) and water (100 μl of total reaction volume) were added to the solution mixture. The peptide coupling reaction was incubated at 25 °C for 24 h. The crude reaction mixtures were analysed by HPLC and MALDI-TOF mass spectrometry.

### General method for the urea cleavage reactions

The hairpin-type intermediate (0.5 nmol) was diluted with MES buffer pH 6 or acetate buffer pH 4 (12.5 μl from a 400 mM aqueous solution), NaCl (5 μl from a 1 M aqueous solution) and water (50 μl of total reacion volume). The urea cleavage reaction was incubated at 60–90 °C at different time intervals. The crude reaction mixtures were analysed by HPLC and MALDI-TOF mass spectrometry.

## Data availability

The data that support the findings of this study are available within the paper and its Supplementary Information.

**Acknowledgements** We thank the Deutsche Forschungsgemeinschaft for supporting this research through the DFG grants: CA275/11-3 (ID: 326039064), CRC1309 (ID: 325871075, A4), CRC1032 (ID: 201269156, A5) and CRC1361 (ID: 393547839, P2). We thank the Volkswagen Foundation for funding this research (grant EvoRib). This project has received funding from the European Research Council (ERC) under the European Union's Horizon 2020 research and innovation programme under grant agreement no. 741912 (EPiR) and under the Marie Skłodowska-Curie grant agreement no. 861381 (Nature-ETN). L.E. thanks the Alexander von Humboldt Foundation for a postdoctoral fellowship (ESP 1214218 HFST-P).

**Author contributions** F.M., L.E., F.X. and E.W. synthesized the modified phosphoramidites and RNA strands and performed the peptide coupling and urea cleavage experiments. M.N. synthesized RNA donor strands and performed preliminary experiments. T.A. refined and developed mechanistic concepts and performed initial proof-of-principle studies. C.-Y.C. and A.P. synthesized modified phosphoramidites. T.C. conceived the project and directed the research. All authors contributed to the analysis of the results and writing of the manuscript.

**Competing interests** The authors declare no competing interests.

**Additional information**
**Correspondence and requests for materials** should be addressed to Thomas Carell.

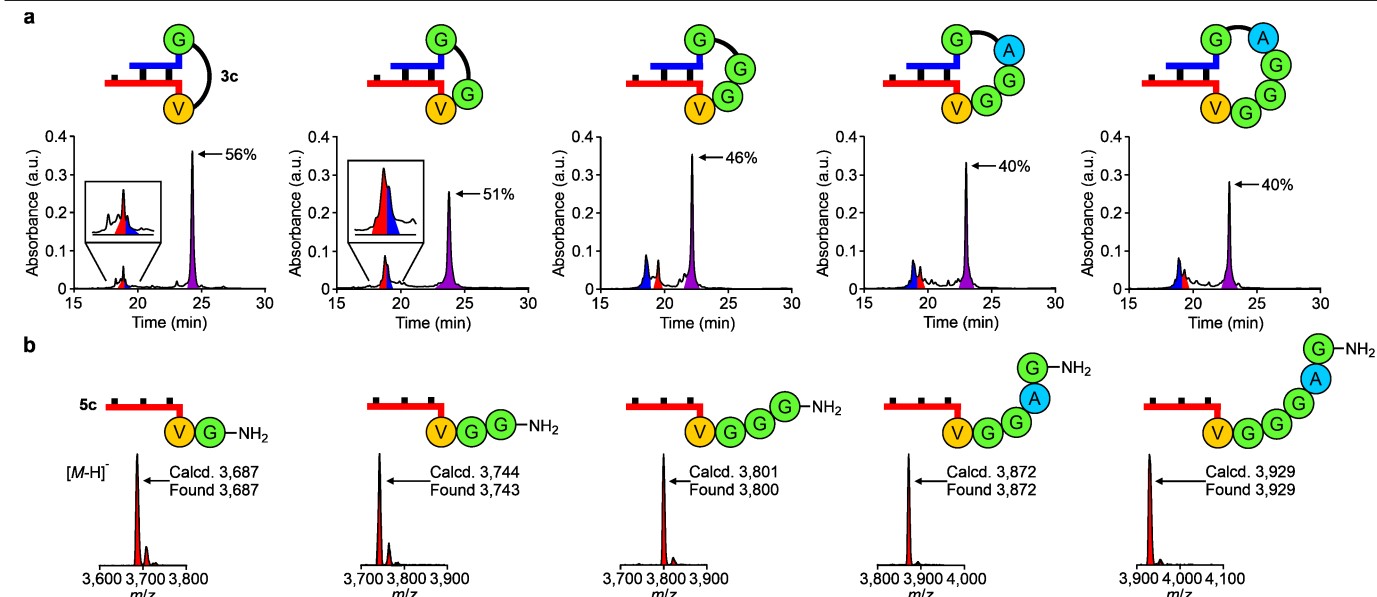

**Extended Data Fig. 1 | Analytical data of the growth of longer peptides on RNA. a**, HPLC chromatograms show the crude mixtures of the coupling reactions (100 mM MES buffer pH 6, 100 mM NaCl, 50 mM EDC/Sulfo-NHS, 25 °C, 24 h) between 5′-m⁶g⁶A-RNA-3′ **1a** and RNA-peptide acceptor strands. **b**, MALDI-TOF mass spectra (negative mode) are shown for the isolated products obtained after the cleavage reactions (100 mM acetate buffer pH 4, 100 mM NaCl, 90 °C, 6 h) of the coupled compounds. In the HPLCs, the RNA strands are coloured: donor in blue; acceptor in red and hairpin-type intermediate in purple.

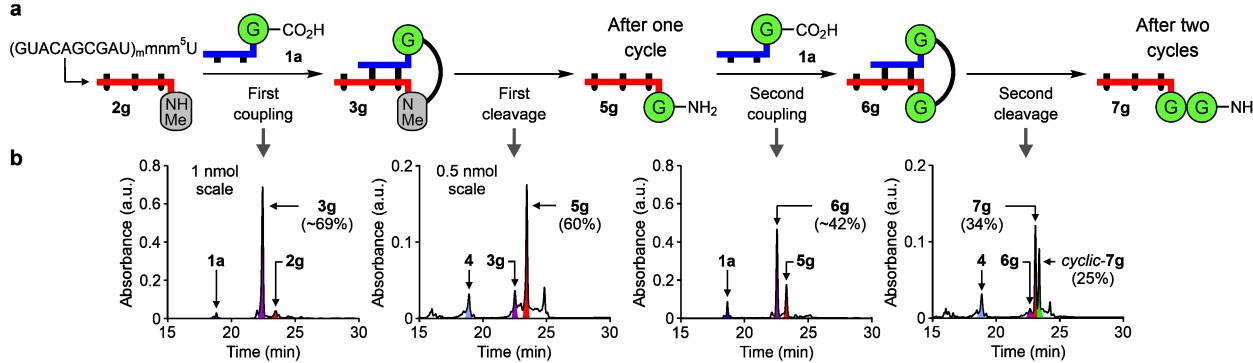

**Extended Data Fig. 2 | RNA-peptide synthesis cycles using a 2′-OMe acceptor strand. a**, Two RNA-peptide synthesis cycles in which the product of each step was separated and added into the next reaction (coupling conditions: 100 mM MES buffer pH 6, 100 mM NaCl, 50 mM DMTMM•Cl, 25 °C, 24 h; cleavage conditions: 100 mM acetate buffer pH 4, 100 mM NaCl, 90 °C, 24 h). **b**, HPLC chromatograms show the crude mixtures of the coupling and cleavage reactions. In the HPLCs, peaks of RNA strands are coloured as in the reaction scheme. The product 3′-ggmnm⁵U-RNA-5′ **7g** was obtained in ≈ 6% overall yield.