## [Peer Review File · Nature]

Manuscript Title: A prebiotically plausible scenario of an RNA-peptide world

Reviewer Comments & Author Rebuttals

Reviewer Reports on the Initial Version:

Referees' comments:

Referee #1 (Remarks to the Author):

The manuscript by Carell and coauthors describes a novel method for RNA-templated peptide synthesis, based on non-canonical nucleotides. The results are supported by an incredible number of supporting experiments and are likely to be of interest for chemists and biochemists, even outside the prebiotic chemistry community. However, I have some concerns regarding the clarity and quality of presentation, and the novelty of the findings.

A recent paper published in Nature Chemistry by Clemens Richert's group describes an alternative method for RNA-templated peptide synthesis, based on single nucleotide-transfer. A similar approach, which explores RNA templated amino acyl transfer (but not peptide synthesis), was recently published by John Sutherland's group in JACS. I understand that these two papers were not published at the time of submission, making it impossible for the authors to comment on them. However, as those papers heavily affect the novelty of this manuscript (in my opinion), an explanation on how the method described in this manuscript would be more advantageous or relevant in a prebiotic context needs to be included in the main text.

The requirement for non-canonical nucleotides to achieve peptide synthesis is an interesting and novel concept. However, the authors should discuss how non-canonical nucleotides and urea-linkages progressively became less relevant for RNA-directed peptide synthesis, considering that modern translation processes is based on different functional groups and chemical reactions.

Also, one could imagine that in an RNA world, oligonucleotides could be involved in both prebiotic replication and translation processes. However, based on the conditions described in the paper, cleavage processes are not as efficient as one would hope, and yield a peptide-bridged dioligonucleotide system as an end-product, incapable of replication or (further) translation. In their last Results paragraph, the authors discuss a different method to achieve urea cleavage, which seem to be quite promising. As this seems to be quite a relevant result to avoid useless end-products and could potentially account for multiple rounds of peptide growth, I believe the author should emphasise more this result and discuss it earlier on in the manuscript.

In my opinion the way the manuscript is written is a bit confusing and misleading. In general, Results are clearly reported, but not discussed - and the Conclusions paragraph does not include a proper discussion of the results. For example, why is the 2'-OMe modification required for better cleavage?

Is base pairing is the only driving force that drives the peptide synthesis process, or there is amino acid selectivity (the authors use leucine, glycine or valine in their parallel synthesis experiments - would they expect the same results regardless of the amino acid loaded on the complementary strand?)?

I also found the Conclusions paragraph quite vague. I believe the relevance of prebiotic RNA-driven peptide synthesis is clear to many - but why is this specific method, involving non-canonical nucleotides, a relevant method for the origins of translation processes? Why are the results relevant in a prebiotic context? And how can they be explained? The Conclusions should, in my opinion, be rewritten to be more focused on the reported results, which need to be thoroughly discussed.

The fragment condensation experiments are quite interesting. Have the authors noticed any solubility issues, since the peptides of choice are not particularly polar? Also, what would happen if polar or charged amino acids were used, instead of glycine/valine? It would be important to know if the efficiency of the loading, coupling or cleavage steps is influenced by the nature of the transferred amino acid/peptide.

I found the “Stepwise growth of peptides” paragraph quite unclear and confusing. Reading the title, I assumed the peptide growth process could be iterated few times to sequentially transfer different amino acids and build longer peptides. However, the short paragraph and the SI seem to point to a different kind of experiment (and the figure doesn’t show any iteration/cyclic process). By reading the text, it seems that the authors performed parallel experiments in which the product of one experiment is separately synthesised and used as substrate for the next experiment. However, the title suggest that the experiments are performed sequentially, rather than in parallel. In order to call this a “stepwise growth of peptides”, the authors should show that the coupling/cleavage reaction can be iterated. This would seem difficult to achieve, considering the low yields of cleavage. However, the high cleavage yields reported in their last Results paragraph should be sufficient to allow for the detection of peptides resulting from iterative cycles of coupling and cleavage.

Overall, I believe the authors deserve a chance to address the novelty concerns related to the recent publication of relatively similar papers, and revise their manuscript to improve the clarity and expand the discussion of their findings.

I believe the most interesting aspects of the paper (which, if fully demonstrated, would still be novel regardless the Richert and Sutherland papers) are the potential for stepwise growth of peptides and the multiple and parallel peptide synthesis (Fig. 5). However, the authors must perform additional experiments to demonstrate that iterative cycles of coupling and cleavage are possible (otherwise there is no stepwise peptide growth, as claimed).

Additionally, it would be relevant to assess if the amino acid nature (polarity, charge, hydrophobicity, aromaticity) plays any role in the efficiency of the coupling, and whether multiple parallel peptide synthesis can be regulated by it.

Referee #2 (Remarks to the Author):

Carell and coworkers describe a novel chemical system in which amino acids, linked to adenine and uracil nucleobases through exocyclic modifications, are condensed into peptides. The modified adenine and uracil bases used in this study are same modified nucleobases that appear in extant tRNAs. With these bases connected to complimentary RNA oligonucleotides, and with the aid of chemical condensing agents, it is shown that single amino acids can be added to a growing peptide chain. The intermediate in this reaction scheme is an RNA hairpin with the loop being the growing peptide. The work is inspired by the question of how protein synthesis was carried out in the hypothetical RNA World. The authors propose that the system described represents a plausible answer to this question.

The chemical system presented is very interesting. There are aspects of this system that are certainly relevant to the RNA World hypothesis. In particular, the used of modified tRNA nucleobases. However, there are shortcomings of the system and apparent discontinuities if one is to accept that this system evolved into extant protein synthesis. These shortcomings raise serious concerns regarding the true relevance of this system to the origin of biological protein synthesis. Without a compelling connection to the historical emergence of extant coded protein synthesis, the value of the system presented may be more in the general realm of novel nucleic acid templated reactions, such as described in references 34 and 39 (Gartner, et al. 2002 and Usanov, 2018).

Major Critiques:

1. The presentation of results in this manuscript initially gave me the impression that the authors had developed a system that achieved continuous, stepwise peptide synthesis. There is even a subsection with the title "Stepwise growth of peptides on RNA." However, this is not the case. Each reaction shown in Figure 3 is set up with starting materials (RNA-peptide conjugates) that are prepared by synthetic methods, not from the products of the previous reaction in the proposed multi-step peptide synthesis scheme. With different reaction conditions being required for the peptide elongation step and the peptide release step, it is not clear how compatible these reactions are and how likely it is that such a stepwise system could have produced peptides in the RNA World. Additionally, the yields of each step are far from quantitative, so the hexapeptides (the last structure shown in Fig. 3) would probably be produced in less than 0.002% in a continuous or one-pot system (based on the yields reported in the figure). The lack of a demonstration of truly progressive system with actual yields tempers my enthusiasm for the system, at this stage of development.

2. While I am intrigued by the use of modified tRNA bases in a scheme for pre-ribosomal peptide synthesis, there are chemical differences between the system presented and extant ribosomal peptide synthesis that seem to be significant discontinuities. For example:

i) As described in the manuscript, in extant ribosomal peptide synthesis the amino acid being added to a growing chain is linked to tRNA through an ester linkage that involves the carboxylic acid group of the amino acid and an OH of ribose (of the tRNA). This linkage serves to bring the amino acid into place and as activation for peptide bond formation during the subsequent step of amino acid addition. Importantly, the extant mechanism allows peptide bond formation and release of an amino acid from its tRNA in one reaction. In the system presented the incoming amino acid is linked through its amino group, which necessitates an additional step for amino acid release after coupling

to the growing peptide. I am not convinced that the mode of amino acid attachment to the nucleobases represents a simpler, a more advantageous, or a more plausibly prebiotic chemistry than what is used in extant life, i.e., activation by esterification with a sugar, which looks plausibly prebiotic.

ii) A related challenge that arises from not using an ester for attachment of the amino acids to the RNA, is the chemical activation that is needed for each amino acid addition. The authors argue that the chemical activating agents they use are plausibly prebiotic, but I think that their reliance on activation of the incoming amino acid with a condensing agent presents an additional challenge for the system to run continuously (rather than in a piece-wise demonstration).

iii) In extant life the mechanism by which each an amino acid is added to a growing chain allows the point of addition to always be the same. That is, between the C-terminal end of the growing peptide and the amino group of the next amino acid. In contrast, in the system presented the growth of the peptide chain is in the opposite direction from that of extant life, and with each amino acid addition the peptide to which amino acids are being added becomes longer in length. The authors show that the system will work up to at least a hexapeptide. Nevertheless, the system shown will likely be limited to relatively short peptides, and how a transition could have transpired to the extant system (with synthesis going in the other direction) is not at all obvious.

Minor critiques.

1) The introduction or discussion sections could include additional references on nucleic acid-amino acid reactions (Berg, J. Biol. Chem., 1958) or using templating chemistry of RNA to drive peptide bond formation (Tamura & Schimmel, PNAS, 2001 & 2003; Yarus, PNAS, 2011). A comparison to such previous works may help illuminate the improvement to such chemistry that is provided by the current this work.

2) Starting on Line 157, the experiment with varying lengths of donor strand that is referenced to Figure 5B is unclear. The diagram appears to show the reactive groups are close in both the 7mer and 11mer complexes. Comparison with Figure 5A and 5B suggests the blue strand is long enough to bind and react. Why it fails to react is not clear. Are the reactive groups not brought into proximity in a thermally stable duplex or are the corresponding duplexes thermally unstable at the reaction temperature? If so, this should be stated. There is a similar problem with base pair mismatch experiment discussed next. For both of these experiments, the SI is not referenced, so it is difficult for the reader to locate sufficient information to fully understand these experiments.

Referee #3 (Remarks to the Author):

As someone who was periodically emphasising (on paper since 2003, then every 3-4 years until now) the decisive importance of understanding the evolution of ribosomal translation through an experimental reconstruction of a biomimetic yet enzyme-free RNA-directed peptide synthesis, I cannot help being very very pleased, to say the least, that it took about 20 years to finally accomplish this formidable task by realising peptides made enzyme- and ribozyme-free by and on RNA in a nucleotide sequence-specific fashion. This pioneering work definitely merits publication in Nature.

It is the second study that demonstrates the experimental feasibility of producing oligopeptides (of an impressive length) from amino acids (and short oligopeptides) that are covalently linked to RNA. The first work—published by the Richert group in *Nature Chemistry* only a few weeks ago, so it should be cited and mentioned in this work—produces much shorter peptides from aminoacylated mononucleotide esters and carboxymethylamino RNA phosphoramidates where, in a certain sense like in this work, each amino acid is attached to either the 3'-terminal or the 5'-terminal (oligo)nucleotide, thus, each partner exposes either its amino terminus or, respectively, its carboxy terminus and the peptide coupling between both ends is achieved by a common water-compatible peptide coupling agent. Having said this, all the rest of the concept and constructs is different in Carell's system.

Richert's approach is a consequent extension of their own work on peptido RNA, to which hydrolytically quite labile aminoacyl 2'/3'-esters were added as amino acid/peptide 'acceptors' (nucleophilic partner in the peptide formation). Consequently, the release of the carboxy terminus of the product peptide and, alas!, also from the starting material, is spontaneous even at low temperatures and neutral pH, and the release of the amino terminus from the phosphoramidate of the product peptide is, on a geological time scale, almost as 'instantaneous' as the carboxy ester hydrolysis, if not for the slightly lowered pH that encourages phosphoramidate hydrolysis. At a first sight, Richert's approach is structurally more similar to the endogenous ribosomal peptidyl transfer, where both peptide donor and acceptor transfer RNAs are carboxy esters. In a putative prebiotic and necessarily cyclic reaction network (due to the repeated formation of peptide bonds in growing oligopeptides) Richert's system seems much more dynamic in a mild and unchanging environment, but also much less robust in a changing environment, for example, through periodic large temperature and significant pH changes, as should be realistically assumed on the early earth.

And then there is this question about codon length. Richert's codon length is a mononucleotide, hence, its translation into a specific amino acid does not at all 'compress the information' content as in biological nature, which is the whole idea of translation: a long linear polymer of a relatively easy-to-copy low 'digit' number (four in endogenous nucleic acids) is translated and compressed into a much shorter polymer of a growing (by taking up new amino acids and codon assignments) high 'digit' number and therefore catalytic competence (doi.org/10.3390/life9010017). In Carell's approach a hexaplet codon is suggestive, e.g. marked by short black lines between annealed donor RNA and acceptor RNA (one line per triplet) but not further mentioned in the paper, see my later comments. The work is based on own long work on non-canonical ribonucleotides, profits much from Grosjean's and Westhof's way of seeing the evolution of endogenous transfer RNA (cited), in particular, the focus on the modifications in positions 34 and 37, and the choice of 'early' amino acids. The fact that short covalent peptide-RNA conjugates are likely to have played a crucial role in early evolution finds support in Szathmáry's 'coding coenzyme handle' concept (not cited). Experimentally, the synthesis of amide bonds from annealed DNA-linked amino acids forming hairpin conjugates was pioneered by David Liu (cited). A deeper discussion about the consequences in terms of prebiotic scenario and time scales, also on codon lengths, in both Richert's and Carell's chemistries, cannot be included in the paper but most probably should be a welcome addition in a News & Views context.

After having carefully studied both, the main text and the whole supporting information file, my

conclusion is, this work is technically virtually flawless. Experimentally, everything has been carried out in sufficient detail and number of repeats, the calibration (for quantification) and control experiments are convincing, no exaggerated over interpretation of data can be found anywhere, and the manuscript and SI are written and explained in a very concise but totally precise way, the graphics are also very intuitive. All abbreviations and structures that are not explained in the main text can be found in the SI with only little effort (search function in PDF files). Of course, one could ask for more variants, more amino acids, more different RNA sequences, mixed hybrid DNA-RNA sequences ("Krishnamurthy-Sutherland"), 2',3'-regioisomeric RNA ("Szostak"), more plausibly prebiotic condensation reagents, "where do we find these amounts of nitrite?", and so forth. The number of possibilities to be tested grows astronomically in a complex system of linear polymers producing other linear polymers, so a certain number of reasonable possibilities in the sense of a proof-of-concept paper is the only way to publish and proceed. As we know from the main author's latest Science papers, this researcher's 'philosophy' is to go and try "paleochemistry", to show what can chemically possibly work out, and not in the first place—although in the second—what really might have happened on early earth, and how exactly. It is the condensed form of an organic chemist's contribution to the question of the origins of life. There are always several different possibilities to be tested and followed equally seriously. Richert's and Carell's approaches are two of those.

What I am missing:

— Most importantly, the courage of the authors to openly oppose the "RNA world" concept. We can read in the abstract that this work shows how a hypothetical RNA world, being quite rightly defined as a really existing system of "self-replicating RNA molecules" (including also 'cross-replicating' I assume), hence, a concept that has no sufficient experimental support for 30 years of trials since the Nobel Prize for Altmann and Czech, can be 'taken up' by a RNA-peptide world ("developed into"). We read that the action of reaction networks carried out by peptide-RNA molecules "represent an option [...] in the RNA world." We can also read somewhere (quite humbly) that the hydrolytic lability of RNA phosphodiester bonds poses a serious problem to the RNA world hypothesis. We read several times about this putative "hen-and-egg" problem of who comes first, peptides or nucleic acids, but we never read that the chemistry that has been tested in this work could quite frankly replace, annihilate the whole RNA world concept. The boldest statement in this direction is: "It is not implausible to assume that some of the peptides could have gained catalytic properties that helped RNA to replicate." Very cautiously formulated... Of course! The whole driving force of the idea is that peptides made by and linked to RNA allowed for their co-evolution that ultimately led to the emergence of proteic nucleic acid polymerases and helicases (leading to the exponential growth in numbers of nucleic acids) and ribozymic amino acid polymerases called ribosomes (two long RNAs associated with 55-80 small, similarly sized ribosomal proteins, see doi:10.1038/nature22998). Really good, useful and promiscuous catalysts, such as proteins, cannot be generally template-copied to be grown in numbers, and nucleic acids cannot be really good, useful and promiscuous catalysts. Both compound classes needed one another right from the start in a mutual collaboration (once termed 'molecular deal'), a functional 'take-over' of an RNA world by a RNA-peptide world has been shown to be highly unlikely, actually impossible (Peter Wills and Charles Carter Jr), and this is why the RNA world concept is a 'dead parrot' (not just sleeping).

— Any mentioning and brief discussion on what this work means for evolving codons. To my understanding, the work suggests that the length of successful codons is strongly dictated by the annealing properties of RNA strands under given prebiotic conditions. On 'a warm prebiotic morning', at temperatures 30-40 °C, we still need some population of dsRNA that can be used to produce hairpinned peptide intermediates. dsRNA hexamers are known (and shown here) to be just stable enough at these temperatures and salt concentrations. I would have thought that pentamers could possibly worked also, albeit less efficiently. The problem with hexaplet codons is the fidelity of the codon-anticodon interaction, both, in terms of mismatches and looped-out nucleotides resulting in frameshifts. Biology optimised the codon length to triplets, most probably, by flanking these aminoacylated RNAs with more RNA, up to the length of endogenous transfer RNA, and of course by conserving post ribosomal chemical modifications at the wobble position 34 and in the anticodon flanking position 37 of endogenous transfer RNA, which is the main underlying theme in this work.

— The pferdefuß of the whole concept is, as I see it, the release of the peptide from the acceptor RNA, as necessitated for the network to be cyclic, not to loose valuable acceptor RNA strands in concentration. We read about the hydrolytic stability of urea linkages with respect to ester connections, but no mentioning of the hydrolytic stability of amide linkages as in the nm5U connection. How to cleave this bond, by hydrolysis? Apparently, the RNA phosphodiester bonds are weaker than an amino acid-nm5U amide bond! This is a major problem for the RNA-peptide synthesis CYCLE. By the way, one arrow is missing in step 4 of Figure 1B, the one that points to the nm5U-acceptor RNA (in addition to the free m6A-donor RNA). Without this arrow (and peptide release) this is no cycle. Suggestion: Do you think it would be possible to cleave this bond under oxidative conditions, like the oxidative cleavage of a (substituted) benzyl group? Such a cleavage would not close the cycle, it would merely release the peptide bearing a N-methylamide carboxy terminus and it would furnish the acceptor RNA with an isoorotic acid moiety at its 3'-terminus...

Minor:

Figures S76 and S77: The rule of thumb is that, to determine whether a melting temperature is concentration dependent or independent you need at least a 50-fold concentration difference, 3 to 8 micromolar isn't quite enough. Also, it would be nice to see the normalised melting curves at both (whatever) low and high concentrations.

Author Rebuttals to Initial Comments:

Point-by-point reply

Referee #1 (Remarks to the Author):

1. A recent paper published in Nature Chemistry by Clemens Richert's group describes an alternative method for RNA-templated peptide synthesis, based on single nucleotide-transfer. A similar approach, which explores RNA templated amino acyl transfer (but not peptide synthesis), was recently published by John Sutherland's group in JACS. I understand that these two papers were not published at the time of submission, making it impossible for the authors to comment on them. However, as those papers heavily affect the novelty of this manuscript (in my opinion), an explanation on how the method described in this manuscript would be more advantageous or relevant in a prebiotic context needs to be included in the main text.

Our manuscript targets a completely different question, and this is now discussed in more detail on page 2 in the revised version. The major idea behind our study was to find plausible scenarios for an RNA-peptide world, in which RNA-peptide conjugates underwent evolutionary optimization and not RNA alone. The work of Richert operates conceptually within the pure RNA world idea. They showed that amino acids, connected to the 5'-end of an RNA strand via an artificial phosphoramidate linkage, can react with an activated second amino acid linked by an ester group to the 3'-end of an incoming nucleoside. This is proximity-driven peptide bond formation. In contrast to this work, we use exclusively naturally occurring non-canonical nucleosides and chemical bonding to show that ancient RNA had the potential to decorate itself with peptides. Richert's work is cited now. It was not known to us when we submitted our paper in the first place.

Sutherland and co-workers reported the aminoacyl transfer from an activated amino acid as acyl phosphate mixed anhydride at the 5'-end of a donor RNA strand to a 2'/3'-OH at the 3'-end of an acceptor RNA strand. This work is very different because it targets the evolution of tRNA synthases. We of course include the reference.

The general concept of amino acyl phosphate mixed anhydrides in RNA-templated peptide synthesis was originally reported by Tamura and Schimmel in 2003. We thank the reviewer for pointing this out to us. The work is now cited as well.

2. The requirement for non-canonical nucleotides to achieve peptide synthesis is an interesting and novel concept. However, the authors should discuss how non-canonical nucleotides and urea-linkages progressively became less relevant for RNA-directed peptide synthesis, considering that modern translation processes is based on different functional groups and chemical reactions.

We now describe the role of non-canonical nucleosides and urea-linkages in RNA-directed peptide synthesis in a new "discussion" paragraph at the end of the manuscript (pages 10-11).

3. Also, one could imagine that in an RNA world, oligonucleotides could be involved in both prebiotic replication and translation processes. However, based on the conditions described in the paper, cleavage processes are not as efficient as one would hope, and yield a peptide-bridged dioligonucleotide system as an end-product, incapable of replication or (further) translation. In their last Results paragraph, the authors discuss a different method to achieve urea cleavage, which seem to be quite promising. As this seems to be quite a relevant result to avoid useless end-products and could potentially account for multiple rounds of peptide growth, I believe the author should emphasise more this result and discuss it earlier on in the manuscript.

We now discuss the results obtained with RNA containing 2'-OMe nucleosides earlier in the revised text (page 5).

4. In my opinion the way the manuscript is written is a bit confusing and misleading. In general, Results are clearly reported, but not discussed - and the Conclusions paragraph does not include a proper discussion of the results. For example, why is the 2'-OMe modification required for better cleavage? Is base pairing is the only driving force that drives the peptide synthesis process, or there is amino acid selectivity (the authors use leucine, glycine or valine in their parallel synthesis experiments - would they expect the same results regardless of the amino acid loaded on the complementary strand?)?

I also found the Conclusions paragraph quite vague. I believe the relevance of prebiotic RNA-driven peptide synthesis is clear to many - but why is this specific method, involving non-canonical nucleotides, a relevant method for the origins of translation processes? Why are the results relevant in a prebiotic context? And how can they be explained? The Conclusions should, in my opinion, be rewritten to be more focused on the reported results, which need to be thoroughly discussed.

This point was also raised by reviewer 3 (comment 5). We changed the conclusion paragraph into a "discussion" section (pages 10-11). In this section, we now discuss all the above-mentioned issues in more detail. We were probably too cautious in the first round.

5. The fragment condensation experiments are quite interesting. Have the authors noticed any solubility issues, since the peptides of choice are not particularly polar? Also, what would happen if polar or charged amino acids were used, instead of glycine/valine? It would be important to know if the efficiency of the loading, coupling or cleavage steps is influenced by the nature of the transferred amino acid/peptide.

We did not experience any solubility issues with RNA strands containing hexapeptides or even decapeptides in aqueous buffered solution.

We thank the referee for this idea. We now performed kinetic studies of a small set of amino acids with different side chains (hydrophobic, aromatic, polar, etc.) in the coupling reaction as suggested. Indeed, we see significant differences. We now discuss the obtained results (page 4) and included the data in Fig. 2. In addition, kinetic studies of selected RNA-peptide conjugates and fragment condensation reactions were performed which we included into the Supporting Information (Figure S40) and which we briefly discuss in the revision (page 5).

6. I found the "Stepwise growth of peptides" paragraph quite unclear and confusing. Reading the title, I assumed the peptide growth process could be iterated few times to sequentially transfer different amino acids and build longer peptides. However, the short paragraph and the SI seem to point to a different kind of experiment (and the figure doesn't show any iteration/cyclic process). By reading the text, it seems that the authors performed parallel experiments in which the product of one experiment is separately synthesised and used as substrate for the next experiment. However, the title suggest that the experiments are performed sequentially, rather than in parallel. In order to call this a "stepwise growth of peptides", the authors should show that the coupling/cleavage reaction can be iterated. This would seem difficult to achieve, considering the low yields of cleavage. However, the high cleavage yields reported in their last Results paragraph should be sufficient to allow for the detection of peptides resulting from iterative cycles of coupling and cleavage.

Indeed, the first experiments were performed in parallel to see how far one can theoretically go with peptide growth. To avoid any confusion, we changed the section title "Stepwise growth of peptides on RNA" in the original manuscript into "Growth of longer peptide structures on RNA". We rephrased the first sentences of this section, and we also removed the dotted lines in Fig. 3b, which may have led to the confusion in the original version. Stepwise growth was performed (pages 8-9), and the data were added to Fig. 5. To our delight, we were able to perform this stepwise growth, even in a one-pot fashion with just filtration steps to remove activator.

I believe the most interesting aspects of the paper (which, if fully demonstrated, would still be novel regardless the Richert and Sutherland papers) are the potential for stepwise growth of peptides and the multiple and parallel peptide synthesis (Fig. 5). However, the authors must perform additional experiments to demonstrate that iterative cycles of coupling and cleavage are possible (otherwise there is no stepwise peptide growth, as claimed).

7. Additionally, it would be relevant to assess if the amino acid nature (polarity, charge, hydrophobicity, aromaticity) plays any role in the efficiency of the coupling, and whether multiple parallel peptide synthesis can be regulated by it.

The coupling efficiency of different amino acids was addressed and included in Fig. 2.

Referee #2 (Remarks to the Author):

There are shortcomings of the system and apparent discontinuities if one is to accept that this system evolved into extant protein synthesis. These shortcomings raise serious concerns regarding the true relevance of this system to the origin of biological protein synthesis. Without a compelling connection to the historical emergence of extant coded protein synthesis, the value of the system presented may be more in the general realm of novel nucleic acid templated reactions, such as described in references 34 and 39 (Gartner, et al. 2002 and Usanov, 2018).

This is all true. Our chemistry differs from what we find in extant protein synthesis. But I believe this may not be critical: The result of our study is that naturally occurring vestige nucleosides of a potential ancient RNA world can lead to peptide growth on RNA. This result forces us to consider an RNA-peptide world in which the RNA-peptide chimeras would have strongly increased catalytic capabilities. The idea is to expand the RNA world concept to an RNA-peptide world idea. Without evolutionary optimization, we can grow up to 6-mers and we can form larger peptides by fragment condensation. It is reasonable to assume that RNA-peptide chimeras had a much higher chance to generate catalytically competent structures that can activate (by adenylation) amino acids and transfer them onto the OH groups of ribose to avoid urea cleavage.

Of course, we need to activate the amino acid (or use the nitriles), but even in current biochemistry the amino acids need activation (by adenylation) before the ester is formed. The difference is small, and I believe it is not too farfetched to assume that RNA-peptide chimeras learned to activate the amino acids with ATP. We are currently trying to realize such a scenario in the laboratory.

Personal comment: If we consider how fragile ester bonds are in water even at near neutral pH over time it is hard to imagine that an RNA world in water, potentially surrounded by nucleophile, would be able to create ribosome-type peptide synthesis of large peptides. The first catalysts were likely poor with low coupling yields. It is potentially easier to assume that the first development was the formation of an RNA-peptide world that could have existed under prebiotically plausible conditions. Then, the system may have started to exclude water by either creating hydrophobic pockets or moving into membranes to develop ATP-driven amino acid activation and ester-based peptide chemistry.

Major Critiques:

1. The presentation of results in this manuscript initially gave me the impression that the authors had developed a system that achieved continuous, stepwise peptide synthesis. There is even a subsection with the title "Stepwise growth of peptides on RNA." However, this is not the case. Each reaction shown in Figure 3 is set up with starting materials (RNA-peptide conjugates) that are prepared by synthetic methods, not from the products of the previous reaction in the proposed multi-step peptide synthesis scheme. With different reaction conditions being required for the peptide elongation step and the peptide release step, it is not clear how compatible these reactions are and how likely it is that such a stepwise system could have produced peptides in the RNA World. Additionally, the yields of each step are far from quantitative, so the hexapeptides (the last structure shown in Fig. 3) would probably be produced in less than 0.002% in a continuous or one-pot system (based on the yields reported in the figure). The lack of a demonstration of truly progressive system with actual yields tempers my enthusiasm for the system, at this stage of development.

In the revised version of the manuscript, we discuss the experiments described in Fig. 3 in more detail (page 5). We removed the dotted lines in Fig. 3b to avoid any confusion with a stepwise growth (see also reply to comment 6 of reviewer 1). We now show in addition progressive peptide synthesis, and this even under one-pot reaction conditions that only required filtration steps. The data are depicted in Fig. 5c,d. The results of these experiments are discussed on pages 8-9. Even under these one-pot conditions, we could quickly realise 5 consecutive reactions (3 couplings and 2 cleavages). The observed yields might allow us to go even beyond this, but we stopped at this point. I believe that every initial RNA-based peptide synthesis particle must have been an inefficient catalyst. More complex structures much beyond $n = 6$ may have required fragment coupling.

2. While I am intrigued by the use of modified tRNA bases in a scheme for pre-ribosomal peptide synthesis, there are chemical differences between the system presented and extant ribosomal peptide synthesis that seem to be significant discontinuities. For example:

i) As described in the manuscript, in extant ribosomal peptide synthesis the amino acid being added to a growing chain is linked to tRNA through an ester linkage that involves the carboxylic acid group of the amino acid and an OH of ribose (of the tRNA). This linkage serves to bring the amino acid into place and as activation for peptide bond formation during the subsequent step of amino acid addition. Importantly, the extant mechanism allows peptide bond formation and release of an amino acid from its tRNA in one reaction. In the system presented the incoming amino acid is linked through its amino group, which necessitates an additional step for amino acid release after coupling to the growing peptide. I am not convinced that the mode of amino acid attachment to the nucleobases represents a simpler, a more advantageous, or a more plausibly prebiotic chemistry than what is used in extant life, i.e., activation by esterification with a sugar, which looks plausibly prebiotic.

See comments above and the discussion of new data on page 2 in the revised manuscript.

ii) A related challenge that arises from not using an ester for attachment of the amino acids to the RNA, is the chemical activation that is needed for each amino acid addition. The authors argue that the chemical activating agents they use are plausibly prebiotic, but I think that their reliance on activation of the incoming amino acid with a condensing agent presents an additional challenge for the system to run continuously (rather than in a piece-wise demonstration).

See comment about adenylation above.

iii) In extant life the mechanism by which each an amino acid is added to a growing chain allows the point of addition to always be the same. That is, between the C-terminal end of the growing peptide and the amino group of the next amino acid. In contrast, in the system presented the growth of the peptide chain is in the opposite direction from that of extant life, and with each amino acid addition the peptide to which amino acids are being added becomes longer in length. The authors show that the system will work up to at least a hexapeptide. Nevertheless, the system shown will

likely be limited to relatively short peptides, and how a transition could have transpired to the extant system (with synthesis going in the other direction) is not at all obvious.

An RNA decorated with multiple peptides in different positions could have functioned as a primitive protein. We simply replace the amide backbone by the (stiff) RNA chain (potentially even duplex), which also organizes amino acid side chains in space. This system could have at some point created the needed hydrophobic pocket to activate the amino acids with ATP and transfer the activated amino acid onto the 2'/3'-OH groups. This would be the transition from the RNA-peptide world to contemporary ribosomal peptide synthesis. We mention these arguments in the "discussion" section (pages 10-11) of the revised text. I agree that the modern ribosome is a perfect catalyst. It generates peptide chains in lengths far beyond of what we chemists can do even today. But such an efficient system may likely not have evolved in one step. What we report in our manuscript is the discovery that non-canonical nucleosides that are considered to be vestiges of an early RNA world can create peptide-decorated RNAs. The peptides will be short, but they can get longer by fragment condensation, and we can attach multiple peptides to the RNA. We believe that these experimental arguments are good for formulating an RNA-peptide world concept.

Minor critiques.

1) The introduction or discussion sections could include additional references on nucleic acid-amino acid reactions (Berg, J. Biol. Chem., 1958) or using templating chemistry of RNA to drive peptide bond formation (Tamura & Schimmel, PNAS, 2001 & 2003; Yarus, PNAS, 2011). A comparison to such previous works may help illuminate the improvement to such chemistry that is provided by the current this work.

We thank the reviewer for the literature suggestions. In the revised version of the manuscript, we implemented all the references.

2) Starting on Line 157, the experiment with varying lengths of donor strand that is referenced to Figure 5B is unclear. The diagram appears to show the reactive groups are close in both the 7mer and 11mer complexes. Comparison with Figure 5A and 5B suggests the blue strand is long enough to bind and react. Why it fails to react is not clear. Are the reactive groups not brought into proximity in a thermally stable duplex or are the corresponding duplexes thermally unstable at the reaction temperature? If so, this should be stated. There is a similar problem with base pair mismatch experiment discussed next. For both of these experiments, the SI is not referenced, so it is difficult for the reader to locate sufficient information to fully understand these experiments.

We thank the reviewer for pointing out these observations.

In the case of the donor strands of different length (Fig. 5b, left), the 11-mer donor and the 11-mer acceptor form a thermodynamically more stable duplex (melting temperature $\sim 59^{\circ}\text{C}$) than the one formed between the 7-mer donor and the same acceptor (melting temperature $\sim 30^{\circ}\text{C}$). A similar explanation applies to the experiment with two 7-mer donor strands in which one of them contains two mismatches (Fig. 5b, right). The fully complementary 7-mer donor strand forms the thermodynamically more stable duplex with the 11-mer acceptor strand. In the revised version of the manuscript, we discuss these two experiments in more detail (page 8).

Referee #3 (Remarks to the Author):

1. As someone who was periodically emphasising (on paper since 2003, then every 3-4 years until now) the decisive importance of understanding the evolution of ribosomal translation through an experimental reconstruction of a biomimetic yet enzyme-free RNA-directed peptide synthesis, I cannot help being very very pleased, to say the least, that it took about 20 years to finally accomplish this formidable task by realising peptides made enzyme- and ribozyme-free by and on RNA in a nucleotide sequence-specific fashion. This pioneering work definitely merits publication in Nature.

We are very excited about this statement.

2. It is the second study that demonstrates the experimental feasibility of producing oligopeptides (of an impressive length) from amino acids (and short oligopeptides) that are covalently linked to RNA. The first work—published by the Richert group in Nature

Chemistry only a few weeks ago, so it should be cited and mentioned in this work— produces much shorter peptides from aminoacylated mononucleotide esters and carboxymethylamino RNA phosphoramidates where, in a certain sense like in this work, each amino acid is attached to either the 3'-terminal or the 5'-terminal (oligo)nucleotide, thus, each partner exposes either its amino terminus or, respectively, its carboxy terminus and the peptide coupling between both ends is achieved by a common water-compatible peptide coupling agent. Having said this, all the rest of the concept and constructs is different in Carell's system.

We agree and we now mention and discuss the Richert's results on page 2 of the revised manuscript. The corresponding reference is also included.

3. Richert's approach is a consequent extension of their own work on peptido RNA, to which hydrolytically quite labile aminoacyl 2'/3'-esters were added as amino acid/peptide 'acceptors' (nucleophilic partner in the peptide formation). Consequently, the release of the carboxy terminus of the product peptide and, alas!, also from the starting material, is spontaneous even at low temperatures and neutral pH, and the release of the amino terminus from the phosphoramidate of the product peptide is, on a geological time scale, almost as 'instantaneous' as the carboxy ester hydrolysis, if not for the slightly lowered pH that encourages phosphoramidate hydrolysis. At a first sight, Richert's approach is structurally more similar to the endogenous ribosomal peptidyl transfer, where both peptide donor and acceptor transfer RNAs are carboxy esters. In a putative prebiotic and necessarily cyclic reaction network (due to the repeated formation of peptide bonds in growing oligopeptides) Richert's system seems much more dynamic in a mild and unchanging environment, but also much less robust in a changing environment, for example, through periodic large temperature and significant pH changes, as should be realistically assumed on the early earth.

And then there is this question about codon length. Richert's codon length is a mononucleotide, hence, its translation into a specific amino acid does not at all 'compress the information' content as in biological nature, which is the whole idea of translation: a long linear polymer of a relatively easy-to-copy low 'digit' number (four in endogenous nucleic acids) is translated and compressed into a much shorter polymer of a growing (by taking up new amino acids and codon assignments) high 'digit' number and therefore catalytic competence (doi.org/10.3390/life9010017). In Carell's approach a hexaplet codon is suggestive, e.g. marked by short black lines between annealed donor RNA and acceptor RNA (one line per triplet) but not further mentioned in the paper, see my later comments. The work is based on own long work on non-canonical ribonucleotides, profits much from Grosjean's and Westhof's way of seeing the evolution of endogenous transfer RNA (cited), in particular, the focus on the modifications in positions 34 and 37, and the choice of 'early' amino acids. The fact that short covalent peptide-RNA conjugates are likely to have played a crucial role in early evolution finds support in Szathmáry's 'coding coenzyme handle' concept (not cited).

The Szathmáry's paper is now cited on page 4 in the revised version of the manuscript. We also follow his thoughts in the discussion section.

4. Experimentally, the synthesis of amide bonds from annealed DNA-linked amino acids forming hairpin conjugates was pioneered by David Liu (cited). A deeper discussion about the consequences in terms of prebiotic scenario and time scales, also on codon lengths, in both Richert's and Carell's chemistries, cannot be included in the paper but most probably should be a welcome addition in a News & Views context.

After having carefully studied both, the main text and the whole supporting information file, my conclusion is, this work is technically virtually flawless.

This is a totally exciting statement as well.

Experimentally, everything has been carried out in sufficient detail and number of repeats, the calibration (for quantification) and control experiments are convincing, no exaggerated over interpretation of data can be found anywhere, and the manuscript and SI are written and explained in a very concise but totally precise way, the graphics are also very intuitive. All abbreviations and structures that are not explained in the

main text can be found in the SI with only little effort (search function in PDF files). Of course, one could ask for more variants, more amino acids, more different RNA sequences, mixed hybrid DNA-RNA sequences ("Krishnamurthy-Sutherland"), 2',3'-regioisomeric RNA ("Szostak"), more plausibly prebiotic condensation reagents, "where do we find these amounts of nitrite?", and so forth. The number of possibilities to be tested grows astronomically in a complex system of linear polymers producing other linear polymers, so a certain number of reasonable possibilities in the sense of a proof-of-concept paper is the only way to publish and proceed. As we know from the main author's latest Science papers, this researcher's 'philosophy' is to go and try "paleochemistry", to show what can chemically possibly work out, and not in the first place—although in the second—what really might have happened on early earth, and how exactly. It is the condensed form of an organic chemist's contribution to the question of the origins of life. There are always several different possibilities to be tested and followed equally seriously. Richert's and Carell's approaches are two of those.

What I am missing:

5. Most importantly, the **courage of the authors to openly oppose the "RNA world" concept**. We can read in the abstract that this work shows how a hypothetical RNA world, being quite rightly defined as a really existing system of "self-replicating RNA molecules" (including also 'cross-replicating' I assume), hence, a concept that has no sufficient experimental support for 30 years of trials since the Nobel Prize for Altmann and Czech, can be 'taken up' by a RNA-peptide world ("developed into"). We read that the action of reaction networks carried out by peptide-RNA molecules "represent an option [...] in the RNA world." We can also read somewhere (quite humbly) that the hydrolytic lability of RNA phosphodiester bonds poses a serious problem to the RNA world hypothesis. We read several times about this putative "hen-and-egg" problem of who comes first, peptides or nucleic acids, but we never read that the chemistry that has been tested in this work could quite frankly replace, annihilate the whole RNA world concept. The boldest statement in this direction is: "It is not implausible to assume that some of the peptides could have gained catalytic properties that helped RNA to replicate." Very cautiously formulated... Of course! The whole driving force of the idea is that peptides made by and linked to RNA allowed for their co-evolution that ultimately led to the emergence of proteic nucleic acid polymerases and helicases (leading to the exponential growth in numbers of nucleic acids) and ribozymic amino acid polymerases called ribosomes (two long RNAs associated with 55-80 small, similarly sized ribosomal proteins, see doi:10.1038/nature22998). Really good, useful and promiscuous catalysts, such as proteins, cannot be generally template-copied to be grown in numbers, and nucleic acids cannot be really good, useful and promiscuous catalysts. Both compound classes needed one another right from the start in a mutual collaboration (once termed 'molecular deal'), a functional 'take-over' of an RNA world by a RNA-peptide world has been shown to be highly unlikely, actually impossible (Peter Wills and Charles Carter Jr), and this is why the RNA world concept is a 'dead parrot' (not just sleeping).

This reviewer is totally correct. We indeed tried to be as cautious as possible, and we potentially went a bit too far with this. In the revised manuscript, we now rewrote the section "conclusion" and fused it with a "discussion" section (pages 10-11) where we put our results into context.

6. Any mentioning and brief discussion on what this work means for evolving codons. To my understanding, the work suggests that the length of successful codons is strongly dictated by the annealing properties of RNA strands under given prebiotic conditions. On 'a warm prebiotic morning', at temperatures 30-40 °C, we still need some population of dsRNA that can be used to produce hairpinned peptide intermediates. dsRNA hexamers are known (and shown here) to be just stable enough at these temperatures and salt concentrations. I would have thought that pentamers could possibly worked also, albeit less efficiently. The problem with hexaplet codons is the fidelity of the codon-anticodon interaction, both, in terms of mismatches and looped-out nucleotides resulting in frameshifts. Biology optimised the codon length to triplets, most probably, by flanking these aminoacylated RNAs with more RNA, up to the length of endogenous transfer RNA, and of course by conserving postribosomal chemical

modifications at the wobble position 34 and in the anticodon flanking position 37 of endogenous transfer RNA, which is the main underlying theme in this work.

To address this point, we reduced the size of the RNA donor strand in new experiments from a total of 7 bases to 5 and even 3 (!) templating bases. We indeed observed peptide coupling in all these cases but with decreasing efficiency. It seems that 3 templating bases is the limit in our system, which would fit nicely to the codon size we see today. The new data were included and discussed on page 4.

7. The pferdefuß of the whole concept is, as I see it, the release of the peptide from the acceptor RNA, as necessitated for the network to be cyclic, not to loose valuable acceptor RNA strands in concentration. We read about the hydrolytic stability of urea linkages with respect to ester connections, but no mentioning of the hydrolytic stability of amide linkages as in the mnm⁵U connection. How to cleave this bond, by hydrolysis? Apparently, the RNA phosphodiester bonds are weaker than an amino acid-mnm⁵U amide bond! This is a major problem for the RNA-peptide synthesis CYCLE. By the way, one arrow is missing in step 4 of Figure 1B, the one that points to the mnm⁵U-acceptor RNA (in addition to the free m⁶A-donor RNA). Without this arrow (and peptide release) this is no cycle. Suggestion: Do you think it would be possible to cleave this bond under oxidative conditions, like the oxidative cleavage of a (substituted) benzyl group? Such a cleavage would not close the cycle, it would merely release the peptide bearing a N-methylamide carboxy terminus and it would furnish the acceptor RNA with an isoorotic acid moiety at its 3'-terminus...

All our attempts to perform a controlled release of a peptide connected to an RNA strand through a mnm⁵U nucleobase were so far unsuccessful. However, full hydrolysis releases the peptide-mnm⁵U under degradation of the RNA strand. The cleavage of the mnm⁵U-peptide bond is currently under heavy investigation in our laboratory.

In the revised version of the manuscript, we describe the results obtained for two/three RNA-peptide synthesis cycles (Fig. 5c,d). We are excited that this could be performed under one-pot conditions (with only intermediate filtrations). We also modified Fig. 1b to show the closed cycle and the elongation of a peptide attached to the RNA acceptor strand.

Minor:

8. Figures S76 and S77: The rule of thumb is that, to determine whether a melting temperature is concentration dependent or independent you need at least a 50-fold concentration difference, 3 to 8 micromolar isn't quite enough. Also, it would be nice to see the normalised melting curves at both (whatever) low and high concentrations.

The melting curves of the hairpin-type intermediates in our work show a behaviour similar to that reported in the literature for self-complementary RNA/DNA strands capable of forming hairpins. Some of the papers used to analyze our results are listed in the Supporting Information: 13. Senior, M. M., Jones, R. A. & Breslauer, K. J. Influence of loop residues on the relative stabilities of DNA hairpin structures. *Proc. Natl. Acad. Sci. U. S. A.* 85, 6242-6246 (1988) and 14. Xodo, L. E., Manzini, G., Quadrifoglio, F., Marel, G. v. d. & van Boom, J. H. Hairpin structures in synthetic oligodeoxynucleotides: sequence effects on the duplex-to-hairpin transition. *Biochimie* 71, 793-803 (1989).

As suggested by this reviewer, we normalized the melting curves of a hairpin-type intermediate. The results are depicted in Figure S87.

We thank all reviewers for their help.

Reviewer Reports on the First Revision:

Referees' comments:

Referee #1 (Remarks to the Author):

I am really excited by the authors' new results, which address my previously discussed main concerns. Now the authors show the stepwise growth of peptides on RNA strands and the study of a broader library of amino acids being loaded and transferred from the donor strand to the acceptor strand.

In line with Reviewer 3, I genuinely appreciate the authors' courage in proposing a new idea based on RNA-peptide chimeras. I am looking forward to seeing how their vision unfolds in the future. I also found particularly intriguing the finding that a 3-mer RNA donor (= codon) is the minimal unit capable of driving the synthesis of such RNA-peptide chimeras.

Overall, I believe this work deserves to be accepted in Nature. However, there are still a few (sometimes major) concerns that the authors should address before acceptance.

- I have a major concern about yield calculation from HPL-chromatograms. The authors state that they use the calibration data obtained for single-strand canonical RNA oligomers. However, the extinction coefficient of ssRNA and dsRNA cannot be assumed to be identical. As such, calibration data for the hairpin-type intermediates would be the appropriate standard to evaluate the concentration of the product, rather than ssRNA standards. Could the authors better clarify how the yield of the product was calculated? How was the extinction coefficient of the hairpin-type intermediate calculated?

- About amino acid diversity: the result showing that Phe had an increased reaction rate compared to other amino acids calls for an (at least hypothetical) explanation. Additionally, the authors included the study of Asp in the SI, which I found particularly interesting. Have the authors used a side chain-protected Asp? If not, do the authors know which carboxylate gets activated? I believe more clarity is needed regarding this experiment in the SI.

- In the caption of Fig. 2, the authors state "HPLC peaks of RNA strands are coloured: Donor in blue; acceptor in red and hairpin-type intermediate in purple". While this is correct for the top HPL-chromatogram, it is not valid for the bottom HPL-chromatogram, where the blue peak is now the "unloaded donor", and the red peak is the product (or "loaded acceptor"). The authors should revise the caption accordingly.

- In Fig. 2d, the authors report two different yields for entry i (Asp). I guess those yields refer to experiments run under different conditions (?). The authors should include this explanation in the caption and do the same in the SI, where the same discrepancy occurs multiple times in several tables' entries.

- The authors frequently mention that peptides could be released upon RNA degradation. What "degradation" process are the authors referring to in this context? Is pH-driven degradation, temperature-driven degradation or chemically-driven degradation? The authors should be more specific about what degradation pathway they are referring to.

When discussing the base-pairing effect, the authors mention that "These results support that full complementarity is needed for efficient peptide synthesis, which establishes the codon-anticodon concept". I believe that a reader, who is less familiar with how the modern biological translation

machinery works, would benefit from a more exhaustive explanation of what the authors mean by this sentence, specifically why full complementarity would establish the codon-anticodon concept, and what the authors mean by “codon-anticodon concept”.

- How can the authors explain that the overall yield in the one-pot experiment is 18% (really cool!) after the second cycle, while the experiment on isolated products is 6%?
- From the same paragraph, I would suggest removing the parallelism between filtration and RNA adsorption on minerals, which IMHO is not needed. If the authors prefer to keep the sentence, they should also explain why only the activator would adsorb onto mineral surfaces, while the acceptor DNA and the incoming new activator would not.
- I appreciate the clarity of the SI. All experimental conditions are clearly reported and explained (other than the calibration experiments on dsRNA that I have already mentioned). My main comment about the SI is that the authors frequently state that the yields are “average of, at least, two experiments”. However, the authors NEVER report an average yield and do not include errors for it in the SI. The authors should revise the SI to include all average and error values.
- For some hairpin-type intermediates, the melting data fit a three-state melting model, and the authors identify two melting temperatures. The authors should explain why some of their hairpin-type intermediates have two melting temperatures and what physico-chemical duplex disassembly process those temperatures are referred to.

Referee #2 (Remarks to the Author):

I have carefully read the revised version of the manuscript by Muller et al., and the response of the authors to each of the three reviews. I am satisfied with the changes made to the manuscript in response to my review. Additionally, I believe that other two reviewers brought up a number of important points that were likewise thoughtfully addressed by the authors. I feel this paper is considerably stronger and now support publication in Nature.

Two minor suggestions:

Line 40: Because the validity of the RNA World is, rightfully, now being questioned in the revised manuscript, the authors may want to change the phrase “the RNA world’ to “the hypothetical RNA world” in the Introduction to read: “Comparative genomics suggests that ribosomal translation is one of the oldest evolutionary processes, that dates back into the hypothetical RNA world.” Just a suggestion.

Line 60: The citation to reference 24 (also by the authors) appears to give credit for the use of wet-dry cycles in prebiotic chemistry to the authors. If there is space, another reference to the use of wet-dry cycles in prebiotic chemistry would be appropriate. For example, Forsythe et al, and Ester-Mediated Amide Bond Formation Driven by Wet–Dry Cycles: A Possible Path to Polypeptides on the Prebiotic Earth, *Angew. Chem. Int. Ed.* 2015, 54, 9871–9875. There are older references, such as by Lahav and Deamer, but this reference shows peptide growth, which is relevant to the current work.

Referee #3 (Remarks to the Author):

About one month ago, a review on the biochemistry, molecular biology and potentially medicinal importance of t6A-processing enzymes and genes has been published in Nucleic Acids Research, Volume 49, Issue 19, 8 November 2021, Pages 10818–10834, by Jonah Beenstock, Frank Sicheri; <https://doi.org/10.1093/nar/gkab865>. Since it is pointing out the general importance of this modification throughout the kingdoms, you might want to cite it in your paper (maybe somewhere near line 72 or 285?)...

It was a very good idea to measure the kinetics of peptide bond formation from different amino acid-RNA donors and acceptors, as well as from different peptide-RNA or/and acceptors. However, the fitting data of the former (as in Fig. S39) is flawed and unacceptable in the present form. It is easy to see by the naked eye that the fitting is far from what one should expect, and it would be evident if the authors had added the corresponding R^2 values. Please add them in the final fittings in Figs. S39 and S40 (in S40 they seem to be acceptable).

Lines 101-103, 134-135 and last column in the table of Fig. 2d: Most importantly, the observed k (small letter k) values must be incorrect for all except that of the slowest donor m6g6A-donor ON1A and the fastest m6f6A ON1g. This fastest reaction, and all the intermediates ones, do level off at some significantly lower strand concentration than 50 μM , which seems to be the reason for the bad fitting quality. I suggest to permit the fitting model to optimise the final concentration as well, or else (second best solution), to diminish by hand the initial concentration to somewhere between 30 and 40 μM depending on the identity and the observed plateau of datapoints after 6 or 8 hours (2 hours for the fastest).

That way, I expect that the difference in the observed peptide coupling rates between that of glycine as donor, and the others, will become much more pronounced. It seems kind of ironic that many of the following fragment couplings have been studied with the slowest donor. Why do you think is m6g6A the slowest? It is the chemically simplest and prebiotically should be the most abundant. A short comment on this would not harm, because in a prebiotic scenario the outcomes would strongly depend on the relative concentrations of donor strands of course.

Lines 106-107: "... the lower limit for productive coupling in our system,..." (and Fig. S78) was carried out at 0 °C and in double strand-stabilising 1 M NaCl. This should be mentioned here, in order to answer the question that one asks oneself: is this lower limit based on the 5 % yield? Not really, or not only, it is also because of the (prebiotic) conditions that cannot be much colder or higher in salinity.

Lines 168-169 and 254-255: Hydantoins are known to form most readily with glycine, for example when activated by carbonyldiimidazole to produce Leuchs' anhydrides. Do the hydantoins also form when the donors are other than m6g6A or m6(peptide)g6A?

Minor:

Line 288: "... small co-evolutionary steps and potentially fragment condensation reactions have

allowed to generate,..."
potential fragment condensation reactions?

Personal comments:

In your first answer to Reviewer 1 you mention that Richert's work includes "an artificial phosphoramidate linkage". What do you mean by artificial, like artwork? Because this group has shown that these phosphoramidates form spontaneously and most rapidly under similarly prebiotic reaction conditions as yours in the present work. If anything, then their aminoacyl acceptors, that were carboxy esters, could be considered being "artificial". Luckily you did not mention this in the revised manuscript.

The general concept of amino acyl phosphate mixed anhydrides in RNA-templated peptide synthesis was originally reported by Hans Kuhn as from section 11 of "Self-organization of molecular systems and evolution of the genetic apparatus" published in *Angewandte Chemie Int. Ed.* 11, 798-811 (1972), and refined it with the actual structures of amino acyl phosphate mixed anhydrides in Kuhn & Waser, *Angew. Chem. Int. Ed.* 20, 500-520 (1981). It might well be that Paul Schimmel had missed out these pioneering works, but when you cite Tamura and Schimmel (rather than Schimmel & Henderson, *PNAS* 91, 11283-11286 (1994))—also Massimo DiGiuglio and Jack Wong—, then it seems only fair to cite the very first pioneer, too.

Author Rebuttals to First Revision:

Point-by-point reply

Referee #1 (Remarks to the Author):

I am really excited by the authors' new results, which address my previously discussed main concerns. Now the authors show the stepwise growth of peptides on RNA strands and the study of a broader library of amino acids being loaded and transferred from the donor strand to the acceptor strand.

In line with Reviewer 3, I genuinely appreciate the authors' courage in proposing a new idea based on RNA-peptide chimeras. I am looking forward to seeing how their vision unfolds in the future. I also found particularly intriguing the finding that a 3-mer RNA donor (= codon) is the minimal unit capable of driving the synthesis of such RNA-peptide chimeras. Overall, I believe this work deserves to be accepted in Nature. However, there are still a few (sometimes major) concerns that the authors should address before acceptance.

1. I have a major concern about yield calculation from HPL-chromatograms. The authors state that they use the calibration data obtained for single-strand canonical RNA oligomers. However, the extinction coefficient of ssRNA and dsRNA cannot be assumed to be identical. As such, calibration data for the hairpin-type intermediates would be the appropriate standard to evaluate the concentration of the product, rather than ssRNA standards. Could the authors better clarify how the yield of the product was calculated? How was the extinction coefficient of the hairpin-type intermediate calculated?

In response to this remark, we measured an additional calibration curve with a hairpin-type intermediate (**ON3a**). The calibration data was reported on Page S29 of the revised SI. The calibration curve of **ON3a** was very similar to the one of the canonical analogue **CON3**.

We explained in more detail on Page S25 that the yields of the hairpin-type products were calculated by integration of the HPLC peaks and the use of the calibration curves. Importantly, the HPLC signals are sharp and not affected by temperature (from 30 to 60°C) suggesting that the reference oligo and the reaction products were likely completely unfolded during the measurements. In addition, we included in Table S6 the extinction coefficients of the oligonucleotides used for the calculations to give full transparency about how the yields were determined.

2. About amino acid diversity: the result showing that Phe had an increased reaction rate compared to other amino acids calls for an (at least hypothetical) explanation. Additionally, the authors included the study of Asp in the SI, which I found particularly interesting. Have the authors used a side chain-protected Asp? If not, do the authors know which carboxylate gets activated? I believe more clarity is needed regarding this experiment in the SI.

In the revised manuscript, we added in Page 4 a sentence to address the issue of the different kinetics observed for the amino acids.

We did not protect the Asp side chain and that is why we obtained two products in this case. The yields for the two products, product 1 and product 2, were reported in the SI (Figures S22 and S26, Tables S14 and S15). Because of the similar yields and identical molecular weights, we did not assign the HPLC signals. We also clarified this in the caption of Fig. 2.

3. In the caption of Fig. 2, the authors state "HPLC peaks of RNA strands are coloured: Donor in blue; acceptor in red and hairpin-type intermediate in purple". While this is correct for the top HPL-chromatogram, it is not valid for the bottom HPL-chromatogram, where the blue peak is now the "unloaded donor", and the red peak is the product (or "loaded acceptor"). The authors should revise the caption accordingly.

We thank the reviewer for pointing this out. We adjusted the colours.

4. In Fig. 2d, the authors report two different yields for entry i (Asp). I guess those yields refer to experiments run under different conditions (?). The authors should include this explanation in the caption and do the same in the SI, where the same discrepancy occurs multiple times in several tables' entries.

These two yields were for the two reaction products obtained when the Asp reacts with the *alpha*-COOH or the side chain COOH. We clarified this in the legend of Fig. 2 as mentioned above.

5. The authors frequently mention that peptides could be released upon RNA degradation. What "degradation" process are the authors referring to in this context? Is pH-driven degradation, temperature-driven degradation or chemically-driven degradation? The authors should be more specific about what degradation pathway they are referring to.

We observed RNA degradation at various conditions. We addressed this on Page 5 of the revised manuscript.

6. When discussing the base-pairing effect, the authors mention that "These results support that full complementarity is needed for efficient peptide synthesis, which establishes the codon-anticodon concept". I believe that a reader, who is less familiar with how the modern biological translation machinery works, would benefit from a more exhaustive explanation of what the authors mean by this sentence, specifically why full complementarity would establish the codon-anticodon concept, and what the authors mean by "codon-anticodon concept".

As recommended, we gave a more detailed explanation of the codon-anticodon concept on Page 8 of the revised manuscript.

7. How can the authors explain that the overall yield in the one-pot experiment is 18% (really cool!) after the second cycle, while the experiment on isolated products is 6%?

In the one-pot experiment, we did not isolate the product strand after each step, which reduces the yield. No product loss due to HPLC purification occurred. In the step-by-step peptide growth, the amount of the donor strand added was different because it was adapted to that of the isolated RNA acceptor strands. For the one-pot experiment, a constant amount of 15 nmol was used in each coupling step. Consequently, the two experiments were performed under slightly different reaction conditions. We clarified on Page 9 that we used the same amount of donor strand in the one-pot experiment for all coupling steps and that the one-pot conditions avoided purification losses. We also adjusted the SI on Page S68.

8. From the same paragraph, I would suggest removing the parallelism between filtration and RNA adsorption on minerals, which IMHO is not needed. If the authors prefer to keep the sentence, they should also explain why only the activator would adsorb onto mineral surfaces, while the acceptor DNA and the incoming new activator would not.

The sentence was removed.

9. I appreciate the clarity of the SI. All experimental conditions are clearly reported and explained (other than the calibration experiments on dsRNA that I have already mentioned). My main comment about the SI is that the authors frequently state that the yields are "average of, at least, two experiments". However, the authors NEVER report an average yield and do not include errors for it in the SI. The authors should revise the SI to include all average and error values.

In the revised version of the SI, we included the errors of all average yields as suggested by the reviewer. In addition, we changed in the tables "Yield (%)" by "Average Yield \pm Error (%)".

10. For some hairpin-type intermediates, the melting data fit a three-state melting model, and the authors identify two melting temperatures. The authors should explain why some of their hairpin-type intermediates have two melting temperatures and what physico-chemical duplex disassembly process those temperatures are referred to.

This remark addressed the melting point curves reported in the Figures S86, S87 and S88. Here, melting involved two transitions from the 1:1 double strand to the self-pairing hairpin-structure, followed by full melting. In the revised SI, we added a clearer representation of this melting behavior, and we included a paragraph in Page S80 that briefly explains the transitions. We also cited in this paragraph two references that support our observations: Breslauer *et al. Proc. Natl. Acad. Sci. U. S. A.* **85**, 6242-6246 (1988) and van Boom *et al. Biochimie* **71**, 793-803 (1989).

Referee #2 (Remarks to the Author):

I have carefully read the revised version of the manuscript by Muller et al., and the response of the authors to each of the three reviews. I am satisfied with the changes made to the manuscript in response to my review. Additionally, I believe that other two reviewers brought up a number of important points that were likewise thoughtfully addressed by the authors. I feel this paper is considerably stronger and now support publication in Nature.

Two minor suggestions:

1. Line 40: Because the validity of the RNA World is, rightfully, now being questioned in the revised manuscript, the authors may want to change the phrase "the RNA world" to "the hypothetical RNA world" in the Introduction to read: "Comparative genomics suggests that ribosomal translation is one of the oldest evolutionary processes, that dates back into the hypothetical RNA world." Just a suggestion.

As suggested by the reviewer, we added "hypothetical" before "RNA world" in Page 2 of the revised version of the manuscript.

2. Line 60: The citation to reference 24 (also by the authors) appears to give credit for the use of wet-dry cycles in prebiotic chemistry to the authors. If there is space, another reference to the use of wet-dry cycles in prebiotic chemistry would be appropriate. For example, Forsythe et al, and Ester-Mediated Amide Bond Formation Driven by Wet-Dry Cycles: A Possible Path to Polypeptides on the Prebiotic Earth, *Angew. Chem. Int. Ed.* 2015, 54, 9871 –9875. There are older references, such as by Lahav and Deamer, but this reference shows peptide growth, which is relevant to the current work.

We thank the reviewer for the suggestion. In the revised manuscript, we included the reference.

Referee #3 (Remarks to the Author):

1. About one month ago, a review on the biochemistry, molecular biology and potentially medicinal importance of t6A-processing enzymes and genes has been published in *Nucleic Acids Research*, Volume 49, Issue 19, 8 November 2021, Pages 10818–10834, by Jonah Beenstock, Frank Sicheri; <https://doi.org/10.1093/nar/gkab865>. Since it is pointing out the general importance of this modification throughout the kingdoms, you might want to cite it in your paper (maybe somewhere near line 72 or 285?).

In the revised version of the manuscript, we added the suggested reference.

2. It was a very good idea to measure the kinetics of peptide bond formation from different amino acid-RNA donors and acceptors, as well as from different peptide-RNA or/and acceptors. However, the fitting data of the former (as in Fig. S39) is flawed and unacceptable in the present form. It is easy to see by the naked eye that the fitting is far from what one should expect, and it would be evident if the authors had added the corresponding R^2 values. Please add them in the final fittings in Figs. S39 and S40 (in S40 they seem to be acceptable).

As recommended by the reviewer, we included the goodness-of-fit in Tables S19 and S20 of the revised SI. These tables correspond to the data shown in Figures S40 and S41, respectively. We added the goodness-of-fit as the sum of squared residuals (SSR) owing to the use of a nonlinear least-squares method. We also included a sentence on Page S52: "In all cases, the fit of the experimental data was good based on the residual values, reported as sum of squared residuals (SSR), and the visual inspection of the curves".

3. Lines 101-103, 134-135 and last column in the table of Fig. 2d: Most importantly, the observed k (small letter k) values must be incorrect for all except that of the slowest donor m6g6A-donor ON1A and the fastest m6f6A ON1g. This fastest reaction, and all the intermediates ones, do level off at some significantly lower strand concentration than 50 μM , which seems to be the reason for the bad fitting quality. I suggest to permit the fitting model to optimise the final concentration as well, or else (second best solution), to diminish by hand the initial concentration to somewhere between 30 and 40 μM depending on the identity and the observed plateau of datapoints after 6 or 8 hours (2 hours for the fastest). That way, I

expect that the difference in the observed peptide coupling rates between that of glycine as donor, and the others, will become much more pronounced.

We thank the reviewer for pointing this out. We re-analyzed the experimental kinetic data shown with a fitting procedure that was similar to that used by Richert and co-workers (see below). The fit of the data was improved, and this leads to larger differences regarding the rate constants between glycine and the other amino acids.

In the revised version of the SI, we included a sentence on Page S52 to explain the fitting and we cited Richert *et al. Nat. Chem.* **13**, 751-757 (2021). The reported values were adjusted in the text of the manuscript and in Fig. 2d.

4. It seems kind of ironic that many of the following fragment couplings have been studied with the slowest donor. Why do you think is m6g6A the slowest? It is the chemically simplest and prebiotically should be the most abundant. A short comment on this would not harm, because in a prebiotic scenario the outcomes would strongly depend on the relative concentrations of donor strands of course.

On Page 4 of the revised version, we added a sentence in which we pointed towards pre-organization as a likely cause of the effect. We are currently investigating this aspect in more detail.

5. Lines 106-107: "... the lower limit for productive coupling in our system,..." (and Fig. S78) was carried out at 0 °C and in double strand-stabilising 1 M NaCl. This should be mentioned here, in order to answer the question that one asks oneself: is this lower limit based on the 5 % yield? Not really, or not only, it is also because of the (prebiotic) conditions that cannot be much colder or higher in salinity.

As suggested, we adjusted the first paragraph on Page 4 to explain that coupling with a 3-mer RNA donor strand required high salt and low temperature conditions to facilitate base pairing.

6. Lines 168-169 and 254-255: Hydantoins are known to form most readily with glycine, for example when activated by carbonyldiimidazole to produce Leuchs' anhydrides. Do the hydantoins also form when the donors are other than m6g6A or m6(peptide)g6A?

We observed the formation of hydantoins with other RNA donor strands as well.

Minor:

7. Line 288: "... small co-evolutionary steps and potentially fragment condensation reactions have allowed to generate,..." potential fragment condensation reactions?

We changed "potentially" by "potential".

Personal comments:

In your first answer to Reviewer 1 you mention that Richert's work includes "an artificial phosphoramidate linkage". What do you mean by artificial, like artwork? Because this group has shown that these phosphoramidates form spontaneously and most rapidly under similarly prebiotic reaction conditions as yours in the present work. If anything, then their aminoacyl acceptors, that were carboxy esters, could be considered being "artificial". Luckily you did not mention this in the revised manuscript.

8. The general concept of amino acyl phosphate mixed anhydrides in RNA-templated peptide synthesis was originally reported by Hans Kuhn as from section 11 of "Self-organization of molecular systems and evolution of the genetic apparatus" published in *Angewandte Chemie Int. Ed.* **11**, 798-811 (1972), and refined it with the actual structures of amino acyl phosphate mixed anhydrides in Kuhn & Waser, *Angew. Chem. Int. Ed.* **20**, 500-520 (1981). It might well be that Paul Schimmel had missed out these pioneering works, but when you cite Tamura and Schimmel (rather than Schimmel & Henderson, *PNAS* **91**, 11283-11286 (1994))—also Massimo DiGiuglio and Jack Wong—, then it seems only fair to cite the very first pioneer, too.

We thank the reviewer for the suggestions. We cited the indicated literature.

Reviewer Reports on the Second Revision:

Referees' comments:

Referee #1 (Remarks to the Author):

I am happy with the changes made by the authors. Therefore, I fully support the publication of the manuscript in its current form.